# The evolution of transposable elements in *Brachypodium distachyon* is governed by purifying selection, while neutral and adaptive processes play a minor role

Robert Horvath[1]*, Nikolaos Minadakis[1], Yann Bourgeois[2,3], Anne C Roulin[1]

[1]Department of Plant and Microbial Biology, University of Zurich, Zurich, Switzerland; [2]DIADE, University of Montpellier, CIRAD, IRD, Montpellier, France; [3]University of Portsmouth, Portsmouth, United Kingdom

*For correspondence:
robert.horvath@uzh.ch

Competing interest: The authors declare that no competing interests exist.

**Abstract** Understanding how plants adapt to changing environments and the potential contribution of transposable elements (TEs) to this process is a key question in evolutionary genomics. While TEs have recently been put forward as active players in the context of adaptation, few studies have thoroughly investigated their precise role in plant evolution. Here, we used the wild Mediterranean grass *Brachypodium distachyon* as a model species to identify and quantify the forces acting on TEs during the adaptation of this species to various conditions, across its entire geographic range. Using sequencing data from more than 320 natural *B. distachyon* accessions and a suite of population genomics approaches, we reveal that putatively adaptive TE polymorphisms are rare in wild *B. distachyon* populations. After accounting for changes in past TE activity, we show that only a small proportion of TE polymorphisms evolved neutrally (<10%), while the vast majority of them are under moderate purifying selection regardless of their distance to genes. TE polymorphisms should not be ignored when conducting evolutionary studies, as they can be linked to adaptation. However, our study clearly shows that while they have a large potential to cause phenotypic variation in *B. distachyon*, they are not favored during evolution and adaptation over other types of mutations (such as point mutations) in this species.

## eLife assessment

This **valuable** study seeks to disentangle the different selective forces shaping the evolutionary dynamics of transposable elements (TEs) in the wild grass Brachypodium distachyon. Using haplotype-length metrics, and genetic and environmental differentiation tests, the authors present **convincing** evidence that positive selection on TE polymorphisms is rare and that the distribution of TE ages points to purifying selection being the main force acting on TE evolution in this species. This study will be relevant for anyone interested in the role of TEs in evolution and adaptation.

## Introduction

Transposable elements (TEs) are an intrinsic part of eukaryotic genomes and their evolution (*Bhattacharyya et al., 1990*; *Hof et al., 2016*; *Feschotte, 2008*; *Qiu and Köhler, 2020*; *Slotkin and Martienssen, 2007*; *Hollister and Gaut, 2009*; *Xiao et al., 2008*; *Gordon et al., 2017*; *Bennetzen and Kellogg, 1997*; *Vitte and Panaud, 2003*; *Piegu et al., 2006*; *Wendel et al., 2016*). In addition to modulating genome size, the ability of TEs to create genetic diversity through insertion and excision events can lead to new phenotypes on which selection can act. TEs can alter phenotypes through

various mechanisms, including the functional disruption of genes (*Bhattacharyya et al., 1990*; *Hof et al., 2016*), large-scale changes in the regulatory apparatus (*Feschotte, 2008*; *Qiu and Köhler, 2020*), alteration of epigenetic landscapes (*Slotkin and Martienssen, 2007*; *Hollister and Gaut, 2009*), ectopic recombination and structural rearrangements (*Xiao et al., 2008*; *Gordon et al., 2017*). In plants, the dynamics of TE loss and proliferation play a major role in genome evolution (e.g. 9–12). TEs therefore constitute potentially important drivers of plant evolution, both in nature and during domestication (*Lisch, 2013*).

Beyond their influence on genome structure, and given that their transpositional activity can be stress-inducible (for review *Negi et al., 2016*), TEs are often regarded as more likely than classical point mutations to produce the diversity needed for individuals to respond quickly to challenging environments (*Rey et al., 2016*; *Dubin et al., 2018*; *Quadrana et al., 2019*). For instance, punctual TE polymorphisms can lead to gains of fitness and evolve under positive selection (*Hof et al., 2016*; *González et al., 2010*; *Studer et al., 2011*; *Barrón et al., 2014*; *Rishishwar et al., 2018*; *Niu et al., 2019*; *Jiang et al., 2022*). TE polymorphisms can even induce more extreme changes in gene expression than single-nucleotide polymorphisms (SNPs) in plants (*Uzunović et al., 2019*; *Castanera et al., 2023*).

Despite such evidence, whether TE polymorphisms are major contributors to adaptation to changing environments is still debated. Indeed, TE transposition can be disruptive, and purifying selection has been shown to play an important role in TE evolution (e.g. *Bourgeois et al., 2020*; *Stritt et al., 2018*). Based on simulations, it has been suggested that the persistence of TE polymorphisms within a genome without an uncontrolled accumulation, can only be achieved if weak purifying selection is the main force governing TE evolution (*Charlesworth, 1991*; *Charlesworth et al., 1997*; *Charlesworth and Charlesworth, 1983*; *Charlesworth, 1996*). The uncertainty surrounding this important question in evolutionary genomics results from the limited number of studies that comprehensively tested the extent to which selection shapes TE allele frequencies, both in plants (*Jiang et al., 2022*; *Lockton et al., 2008*) and animals (*Bourgeois et al., 2020*; *Boissinot et al., 2006*; *Blumenstiel et al., 2014*; *Rech et al., 2019*; *Mérel et al., 2021*) and characterized the distribution of fitness effects of new TE insertions. To clarify this question, we used the plant model system *Brachypodium distachyon* (*International Brachypodium Initiative, 2010*) to disentangle the effects of purifying and positive selection on TE polymorphisms in natural populations.

*B. distachyon* is a wild annual grass endemic to the Mediterranean basin and Middle East. Recent genetic studies based on more than 320 natural accessions spanning from Spain to Iraq (hereafter referred to as the *B. distachyon* diversity panel) revealed that *B. distachyon* accessions cluster into three main genetic lineages (the A, B, and C genetic lineages), which further divide into five main genetic clades that display little evidence for historical gene flow (*Figure 1A*; *Stritt et al., 2022*; *Minadakis et al., 2023*). Niche modeling analyses suggest that the species moved southward during the last glacial period and recolonized Europe and the Middle East within the last five thousand years (*Minadakis et al., 2023*). Consequently, while some *B. distachyon* genetic clades currently occur in the same broad geographical areas (*Figure 1A*), natural accessions are adapted to a mosaic of habitats (*Stritt et al., 2022*; *Minadakis et al., 2023*). These past and more recent shifts in the species distribution led to clear footprints of positive selection in the genome (*Minadakis et al., 2023*; *Bourgeois et al., 2018*) and make *B. distachyon* an ideal study system to investigate the contribution of TEs to the adaptation of plants in the context of environmental changes.

In *B. distachyon*, TEs are exhaustively annotated and account for approximately 30% of the genome (*International Brachypodium Initiative, 2010*). Recent TE activity has been reported for many families, but despite past independent bottlenecks and expansions experienced by the different genetic clades, no lineage-specific TE family activity has been observed (*Lockton et al., 2008*). Rather, TE activity tends to be homogeneous throughout the species range and across genetic clades, indicating a high level of conservation of the TE regulatory apparatus (*Lockton et al., 2008*). While purifying selection shapes the accumulation patterns of TEs in this species (*Lockton et al., 2008*), some TE polymorphisms have been observed in the vicinity of genes (*Lockton et al., 2008*), potentially affecting gene expression (*Wyler et al., 2020*). These early studies, based on a relatively small number of accessions originating exclusively from Spain and Turkey, suggested that TE polymorphisms could contribute to functional divergence and local adaptation in *B. distachyon* (*Lockton et al., 2008*).

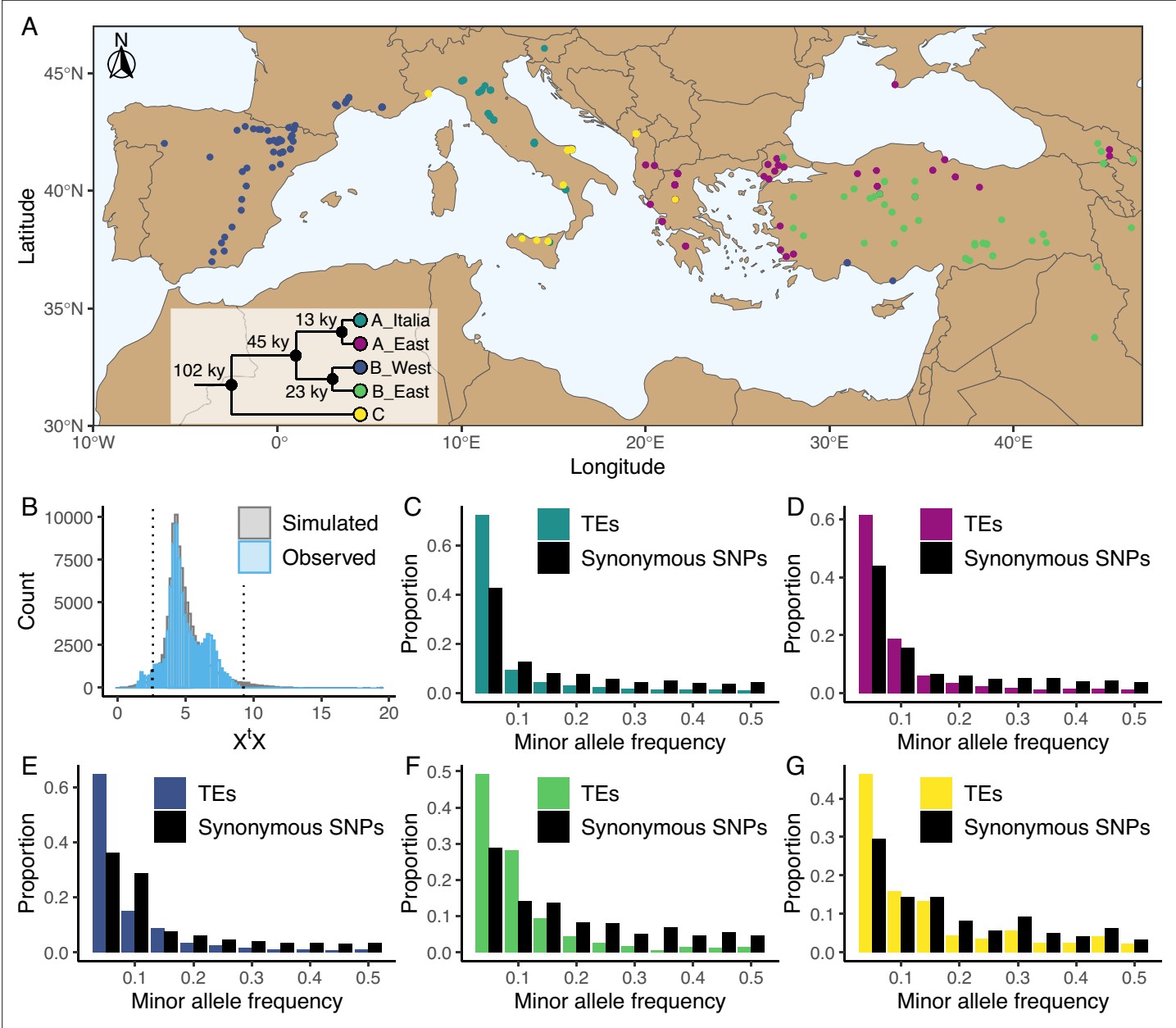

**Figure 1.** Distribution of the studied accessions and TE polymorphism frequencies. (**A**) Map showing the geographical distribution of the accessions (n = 326) used in the current study. The phylogenetic tree illustrates the phylogeny between the five genetic clades. This panel was made based on the data and results published by *Stritt et al., 2022* and *Minadakis et al., 2023*. (**B**) Observed (blue, n = 97,660) and simulated (gray, n = 100,000) X$^t$X values of TE polymorphisms in *B. distachyon*. Dotted lines show the 2.5% and 97.5% quantiles of the simulated X$^t$X values. (**C-G**) Folded site frequency spectrum of TE polymorphisms and synonymous SNPs in all clades. (**C**) A_East (n$_{TE}$ = 37,563; n$_{SNP}$ = 92,130); (**D**) A_Italia (n$_{TE}$ = 32,753; n$_{SNP}$ = 82,101); E: B_West (n$_{TE}$ = 48,315; n$_{SNP}$ = 99,953); F: B_East (n$_{TE}$ = 25,757; n$_{SNP}$ = 60,539); G: C (n$_{TE}$ = 24,161 ; n$_{SNP}$ = 78,681). Principal Component Analyses using TE, SNP, retrotransposon and DNA-transposon are shown in *Figure 1—figure supplements 1 and 2*. Observed correlation between age in generations and frequency of synonymous SNPs in the four derived genetic clades are shown in *Figure 1—figure supplement 3*. Distribution of the observed TE age scaled by the effective population size (N$_e$) in the four derived genetic clades are shown in *Figure 1—figure supplement 4*. Folded site frequency spectrum of DNA-transposons and retrotransposons are shown in *Figure 1—figure supplements 5 and 6*.

The online version of this article includes the following figure supplement(s) for figure 1:

**Figure supplement 1.** Principal Component Analyses using TE (left panel, n = 97,660) and SNP (right panel, n = 182,801) polymorphisms.

**Figure supplement 2.** Principal Component Analyses using retrotransposon (left panel, n = 9,172) and DNA-transposon (right panel, n = 52,249) polymorphisms.

**Figure supplement 3.** Observed correlation between age in generations and frequency of synonymous SNPs in the four derived genetic clades.

*Figure 1 continued on next page*

*Figure 1 continued*

**Figure supplement 4.** Distribution of the observed TE age scaled by the effective population size ($N_e$) in the four derived genetic clades of *B. distachyon*.

**Figure supplement 5.** Folded site frequency spectrum of DNA-transposons and synonymous SNPs in all genetic clades.

**Figure supplement 6.** Folded site frequency spectrum of retrotransposons and synonymous SNPs in all genetic clades.

To test this hypothesis, we used the *B. distachyon* diversity panel to identify TE polymorphisms in a large set of 326 natural accessions spanning the whole species distribution. We combined a set of population genomic analyses to assess the proportion of TE polymorphisms associated with positive or purifying selection as well as neutral evolution. We also quantified the strength of purifying selection through forward simulations. Altogether, our work provides the first quantitative estimate, to our knowledge, of the adaptive, neutral, and disruptive potential of TEs, while accounting for changes in TE activity, in a plant harboring a relatively small genome. Altogether, our result advocate against an extended role of TEs in recent adaptation.

## Results

### Genetic variation in *Brachypodium distachyon*

Using the *B. distachyon* diversity panel (*Figure 1A*), we identified 97,660 TE polymorphisms in our *B. distachyon* dataset, of which 9172 were retrotransposons, 52,249 were DNA-transposons and 36,239 were unclassified. We also identified 9 million SNPs across the 326 samples, including 182,801 synonymous SNPs. A Principal Component Analysis (PCA) performed either with SNPs or TE polymorphisms reflects the previously described population structure of *B. distachyon* (*Stritt et al., 2022*; *Minadakis et al., 2023*), with the first two components of the PCA splitting the data according to the demographic structure (*Figure 1—figure supplement 1*). Investigating the genetic variation caused by retrotransposons and DNA-transposons revealed that the observed diversity in retrotransposons strongly correlated with the demographic structure (Mantel test; $r=0.79$, $p$-value = 0.001), while the observed diversity in DNA-transposons only had a weaker correlation (Mantel test; $r=0.36$, p-value = 0.001) with the demographic structure (*Figure 1—figure supplement 2*).

From the initial TE and SNP dataset, we could estimate the time of origin in generations (age) of 50,891 TE polymorphisms and 108,855 synonymous SNPs based on pairwise differences in identity by descent (IBD) regions around the focal mutation (see Materials and methods). The results of the age estimate analysis were checked by contrasting the observed correlation between allele age and frequency of synonymous SNPs to the theoretical predictions of *Kimura and Ohta, 1973* for neutrally evolving mutations. We found that, the observed correlation matched expectations (*Figure 1—figure supplement 3*), with older alleles found on average at higher frequencies than younger ones. Furthermore, most TE polymorphisms in our dataset were young and only a few were very old (*Figure 1—figure supplement 4*).

**Table 1.** ANCOVA predicting the number of fixed TE polymorphisms per clade in candidate regions under positive selection.

| Variable | Sum of squares | degrees of freedom | F value | p value |
|---|---|---|---|---|
| Total number of TEs in the region | 28969.6 | 1 | 35405.64 | <0.001 |
| TE superfamily | 887.5 | 14 | 77.48 | <0.001 |
| Clade | 587 | 3 | 239.13 | <0.001 |
| Genomic region | 136.7 | 80 | 2.09 | <0.001 |
| TE age | 45.5 | 2 | 27.81 | <0.001 |
| High iHS | 0 | 1 | 0.03 | 0.869 |

**Table 2.** ANCOVA predicting the allele frequency of TE polymorphisms per clade in candidate regions under positive selection.

| Variable | Sum of squares | degrees of freedom | *F* value | p value |
|---|---|---|---|---|
| TE superfamily | 453.2 | 14 | 247.3 | <0.001 |
| Clade | 17.7 | 3 | 45.18 | <0.001 |
| Genomic region | 147 | 80 | 14 | <0.001 |
| TE age | 2 | 2 | 7.7 | <0.001 |
| High iHS | 0.1 | 1 | 0.79 | 0.374 |

## The overall contribution of TEs to clade differentiation and adaptation is limited

To examine the overall contribution of TEs to evolution and adaptation in *B. distachyon*, we first identified regions of the genomes that were likely affected by recent selective sweeps. The fast increase in the frequency of a beneficial allele is expected to lead to a longer than average haplotype around the mutation under positive selection. Such events (known as selective sweeps) can be identified by computing the integrated haplotype score (iHS) around focal mutations (*Voight et al., 2006*). We therefore computed iHS along the genome for the four derived genetic clades. Regions of the genomes with significantly higher iHS than average are expected to harbor mutations that were under positive selection during evolution and adaptation. We hypothesized that if TEs constitute an important part of the genetic makeup that led to adaptation in a given genetic clade, then they should be more frequently fixed or at higher frequencies in regions with high iHS than in the corresponding regions that did not experience recent selective sweeps in other clades.

First, we tested if more TE polymorphisms were fixed in a specific region of the genome if a genetic clade had a high iHS, and presumably experienced a selective sweep, than in other genetic clades. An analysis of covariance (ANCOVA) revealed that the number of fixed TE polymorphisms per clade did not significantly differ between high iHS regions and the same regions in other clades (*Table 1*). These results indicate that there is no correlation between the overall number of fixed TE polymorphisms per clade in a region and recent selective sweeps. However, the number of fixed TEs in genomic regions along the genome was significantly affected by the total number of TEs in the region, the TE superfamily, the TE age, the genetic clade and the overall genetic features of the region (e.g. recombination rate, see Materials and methods) but not by the iHS itself (*Table 1*). Similarly, we tested if the allele frequency of TE polymorphisms was significantly higher in a specific region of the genome if a genetic clade had a high iHS than in other genetic clades. A second ANCOVA revealed that the allele frequency of TE polymorphisms was significantly influenced by the TE superfamily, TE age, clade and overall genetic features of the region but not by the iHS (*Table 2*). These results indicate that TEs in high iHS regions did not experience a significant increase in their frequency and that TEs in high iHS regions are experiencing the same selective constraints as other TEs. Similar results were obtained when investigating the number of fixed TE polymorphisms (*Supplementary file 1 -table 1a*) and the allele frequency of TE polymorphisms (*Supplementary file 1-table 1b*) in high iHS regions using a subset of our dataset with an expected lower false negative TE call rate, that only included samples with a genome-wide mapping coverage of at least 20 x (see Discussion and Materials and ethods for more details).

A complementary approach to explore the impact of positive selection on TEs consists in investigating their genetic differentiation among populations. Using the five genetic clades as focal populations, we computed $X^tX$ values, a standardized measure of genetic differentiation corrected for the neutral covariance structure across populations (*Günther and Coop, 2013*; *Olazcuaga et al., 2020*), for each TE polymorphism. Mutations affected by positive selection are expected to be over-differentiated between clades and display significantly higher $X^tX$ values than other mutations (*Olazcuaga et al., 2020*). In contrast, a low $X^tX$ value implies that the mutation is less differentiated than other mutations and potentially evolves under balancing selection, whereas purifying selection and a neutral evolution are not expected to impact the differentiation of a mutation among populations (*Günther and Coop, 2013*). We contrasted the observed $X^tX$ values computed for each TE polymorphism to a

simulated pseudo-observed dataset (simulated observations under the demographic model inferred from the covariance matrix of the SNP dataset, for more details see *Olazcuaga et al., 2020*) and found that only a small fraction of the TE polymorphisms (0.06%) displayed $X^tX$ values higher than the 97.5% quantile of the simulated values (*Figure 1B*). This indicates that only a few TE polymorphisms are over-differentiated among genetic clades and might have been affected by positive selection. However, a relatively larger portion of the TE polymorphisms (4.3%) displayed $X^tX$ values smaller than the 2.5% quantile of the simulated values (*Figure 1B*), indicating that balancing selection might also shape TE frequency in *B. distachyon*.

To further examine the contribution of TEs to adaptation, we tested whether and how many TE polymorphisms were significantly associated with environmental factors. If the presence of a TE provides an advantage in a certain environment and contributes to adaptation, we expected a correlation between the environment and the presence/absence of this TE. In this context, we performed genome-environment association analyses (GEA) using all TEs and SNPs identified across the 326 samples and 32 environmental factors associated with precipitation, solar radiation, temperature, elevation and aridity (see in Materials and methods for the full list). The GEA revealed that only nine of the 97,660 TE polymorphisms were significantly associated with some environmental factors (*Supplementary file 1-table 1c*), confirming that TEs only had a limited contribution to adaptation in *B. distachyon*. Importantly, two of these nine TEs were found in a gene, and three were in the vicinity of genes (less than 2 kilobase (kb) away, *Supplementary file 1-table 1c*).

## Purifying selection dominates the evolution of TE polymorphisms in *B. distachyon*

To further characterize the forces governing the evolution of TE polymorphisms in *B. distachyon*, we examined the genome-wide frequency distribution of TEs. We first computed the folded site frequency spectrum (SFS) and found that the folded SFS of TE polymorphisms was shifted toward a higher proportion of rare minor alleles compared to neutral sites in all genetic clades (*Figure 1C–G*). Splitting the TE data into DNA-transposons and retrotransposons resulted in similar folded SFS and shifts in both TE classes (*Figure 1—figure supplements 5 and 6*).

These shifts could be the result of purifying selection as the analyses presented above indicate that positive selection has a negligible effect on TE polymorphism frequencies in *B. distachyon*. However, in contrast to SNPs, TEs do not evolve in a clock-like manner, as their transposition rate is known to vary between generations (*García Guerreiro, 2012*; *Belyayev, 2014*). Changes in transposition rate and purifying selection can lead to similar shifts in the SFS but can be disentangled using age-adjusted SFS (*Horvath et al., 2022*). In brief, if TE polymorphisms are evolving neutrally, they are expected to accumulate on average at the same rate in a population as neutral SNPs of the same age. Hence, Δ frequency, the difference between the average frequency of TE polymorphisms and neutral sites in a specific age bin, will remain close to 0 regardless of the polymorphisms' age. In contrast, if TE polymorphisms evolve under purifying selection, they will tend to occur at lower frequencies than neutral SNPs of the same age, as selection will prevent them from accumulating in the population. Consequently, the Δ frequency will reach negative values for older TE polymorphisms (*Horvath et al., 2022*).

Because this model does not allow for back mutations, as typically observed for DNA-transposons that can excise from the genome, we primarily investigated the age-adjusted SFS of retrotransposons in the four derived clades. This analysis revealed that retrotransposons are indeed prevented by natural selection from randomly accumulating, as older retrotransposons are significantly less frequent than neutral SNPs of the same age (*Figure 2*; one-sided Wilcoxon test, Bonferroni corrected p-value <0.01).

As previous studies showed that the distance between TE polymorphisms and the next gene can impact the strength of selection affecting TEs (*Hollister and Gaut, 2009*; *Wright et al., 2003*; *Horvath and Slotte, 2017*), we further split our retrotransposon polymorphisms into three categories based on their distance to the next gene: retrotransposons (i) in and up to 1 kb away from genes, (ii) between 1 kb and 5 kb away and (iii) more than 5 kb away. The age-adjusted SFS of all three categories displayed the same pattern as that observed for the whole retrotransposon polymorphism dataset: older retrotransposon polymorphisms were significantly less frequent than neutral sites of the same age regardless of their distance to genes (one-sided Wilcoxon test, Bonferroni corrected

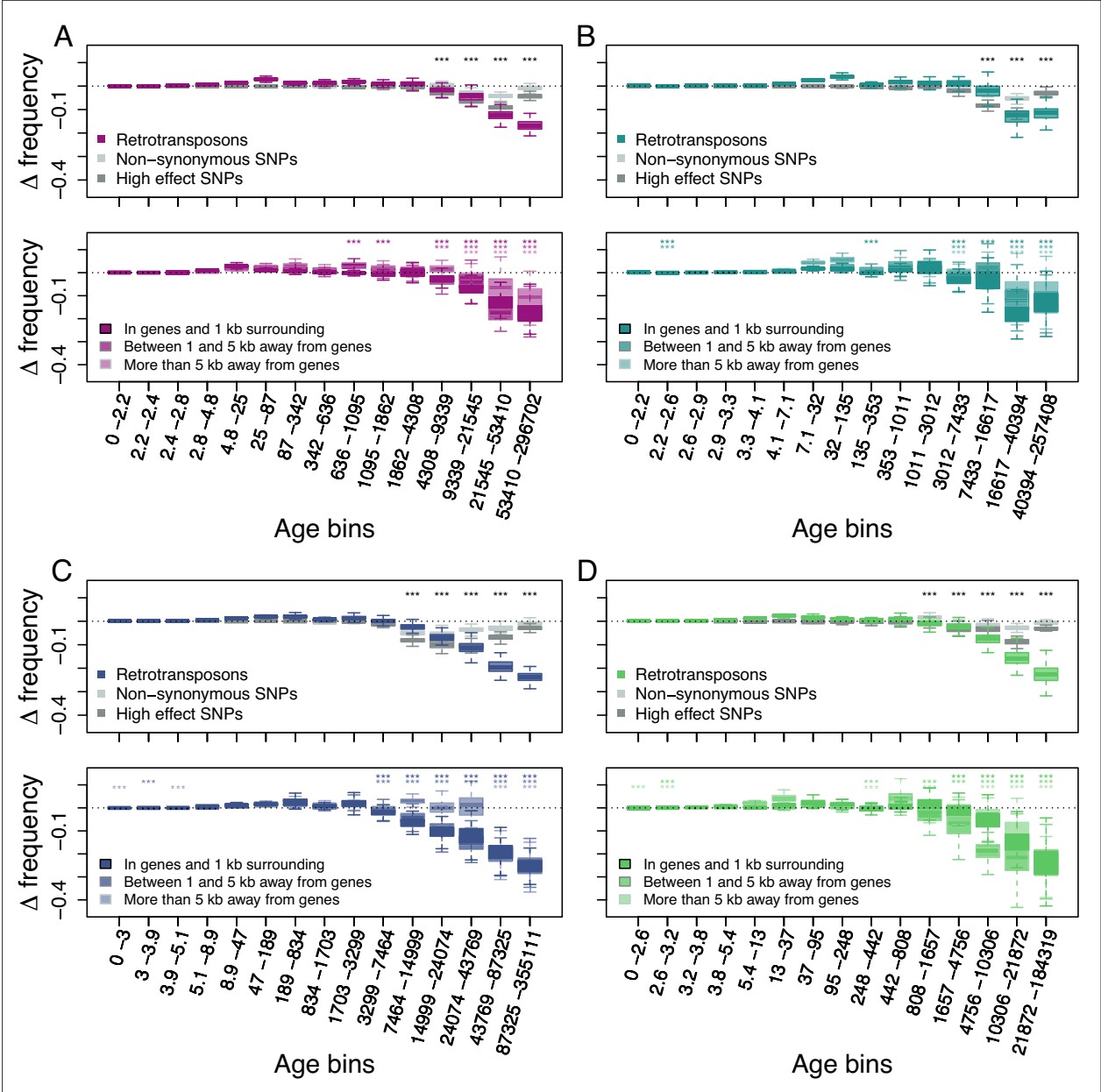

**Figure 2.** Age-adjusted SFS of retrotransposons. The top row shows the age-adjusted SFS of all retrotransposons (colored), non-synonymous SNPs (light gray) and high effect SNPs (dark gray) in the four derived clades. The bottom row shows the age-adjusted SFS of retrotransposons based on their distance to the next gene in the four derived clades. The X axes show the age range of the mutations in each bin, and the age range of each bin was chosen so that each bin represents the same number of retrotransposon observations in the top row. The different columns show the four derived clades: (**A**) A_East ($n_{retrotransposon}$ = 2,106, $n_{non-synonymous\ SNP}$ = 10,000, $n_{high\ effect\ SNP}$ = 9,050, $n_{retrotransposon\ in\ genes\ and\ 1\ kb\ surrounding}$ = 733, $n_{retrotransposon\ between\ 1\ and\ 5\ kb\ away\ from\ genes}$ = 664, $n_{retrotransposon\ more\ than\ 5\ kb\ away\ from\ genes}$ = 709); (**B**) A_Italia ($n_{retrotransposon}$ = 1,232, $n_{non-synonymous\ SNP}$ = 10,000, $n_{high\ effect\ SNP}$ = 7,273, $n_{retrotransposon\ in\ genes\ and\ 1\ kb\ surrounding}$ = 390, $n_{retrotransposon\ between\ 1\ and\ 5\ kb\ away\ from\ genes}$ = 388, $n_{retrotransposon\ more\ than\ 5\ kb\ away\ from\ genes}$ = 454); (**C**) B_West ($n_{retrotransposon}$ = 2,081, $n_{non-synonymous\ SNP}$ = 10,000, $n_{high\ effect\ SNP}$ = 10,000, $n_{retrotransposon\ in\ genes\ and\ 1\ kb\ surrounding}$ = 812, $n_{retrotransposon\ between\ 1\ and\ 5\ kb\ away\ from\ genes}$ = 647, $n_{retrotransposon\ more\ than\ 5\ kb\ away\ from\ genes}$ = 622); (**D**) B_East ($n_{retrotransposon}$ = 1,035, $n_{non-synonymous\ SNP}$ = 10,000, $n_{high\ effect\ SNP}$ = 6,306, $n_{retrotransposon\ in\ genes\ and\ 1\ kb\ surrounding}$ = 387, $n_{retrotransposon\ between\ 1\ and\ 5\ kb\ away\ from\ genes}$ = 311, $n_{retrotransposon\ more\ than\ 5\ kb\ away\ from\ genes}$ = 337). Boxplots are based on 100 estimations of D frequency. Significant deviations of D frequency estimates from 0 in the age-adjusted SFS of retrotransposons are shown with asterisks (one-side Wilcoxon tests, Bonferroni corrected p-value <0.01: ***). Age-adjusted SFS of DNA-transposons are shown in *Figure 2—figure supplement 1*. Age-adjusted SFS of simulated mutations under negative selection in the four derived clades transposons are shown in *Figure 2—figure supplement 2*. Age-adjusted SFS of retrotransposons in accessions with at least 20 x coverage are shown in *Figure 2—figure supplement 3*. Age-adjusted SFS of retrotransposons more than 5 kb away from genes are shown in *Figure 2—figure supplement 4*. Age-adjusted SFS of Copia, Ty3, Helitron and MITE TEs are shown in *Figure 2—figure supplements 5–8*.

The online version of this article includes the following figure supplement(s) for figure 2:

*Figure 2 continued on next page*

*Figure 2 continued*

**Figure supplement 1.** Age-adjusted SFS of DNA-transposons (colored), non-synonymous SNPs (light gray) and high effect SNPs (dark gray) in the four derived clades.

**Figure supplement 2.** Age-adjusted SFS of simulated mutations under negative selection in the four derived clades.

**Figure supplement 3.** Age-adjusted SFS of retrotransposons in accessions with at least 20 x coverage.

**Figure supplement 4.** Age-adjusted SFS of retrotransposons (colored) and SNPs (gray) more than 5 kb away from genes in the four derived clades.

**Figure supplement 5.** Age-adjusted SFS of Copia TEs in the four derived clades.

**Figure supplement 6.** Age-adjusted SFS of Ty3 TEs in the four derived clades.

**Figure supplement 7.** Age-adjusted SFS of Helitron TEs in the four derived clades.

**Figure supplement 8.** Age-adjusted SFS of MITE TEs in the four derived clades.

p-value <0.01), indicating that retrotransposons more than 5 kb away from genes are also affected by purifying selection (*Figure 2*).

Retrotransposon polymorphisms tended to be more deleterious than SNPs predicted to have a high impact on fitness. Indeed, the age-adjusted SFS of retrotransposons resulted in a larger deviation of Δ frequency from 0 than for non-synonymous SNPs and high effect SNPs (*Figure 2*). In addition, Δ frequency in the oldest (last) age bin was significantly more negative than in all other age bins in the A_East, B_East and B_west clades (one-sided Wilcoxon test, Bonferroni corrected p-value <0.01). In the A_Italia clades the oldest age bin was not significantly different from the second oldest age bin (two-sided Wilcoxon test, Bonferroni corrected *p* value N.S.). While older non-synonymous SNPs and high effect SNPs were generally less frequent than neutrally evolving SNPs at the same age, the negative Δ frequency trend was reversed for the oldest non-synonymous SNPs and high effect SNPs (*Figure 2*). In all clades, Δ frequency in the oldest age bin was significantly higher than at least the lowest Δ frequency observed in the other age bins for non-synonymous SNPs, as well as high effect SNPs (one-sided Wilcoxon test, Bonferroni corrected p-value <0.01). This might be because not all predicted non-synonymous SNPs and high effect SNPs might result in fitness effects. Those SNPs can therefore evolve neutrally or nearly neutrally and persist as polymorphic SNPs much longer in a population than those affecting fitness negatively. Hence, even the oldest retrotransposon polymorphisms seem to be mostly non-neutral and are affected by purifying selection.

To assess whether similar forces may drive retrotransposon and DNA-transposon evolution, we repeated the analysis for DNA-transposons. The age-adjusted SFS of DNA-transposons revealed very similar patterns, with Δ frequency showing significant deviations from 0 in older age bins (one-sided Wilcoxon test, Bonferroni corrected p-value <0.01), but DNA-transposon polymorphisms seemed less deleterious than non-synonymous SNPs and high effect SNPs (*Figure 2—figure supplement 1*).

## Forward simulations allow us to quantify the strength of purifying selection

To evaluate to what extent the proportion of neutrally evolving mutations in the focal group of mutations affects the shape of the age-adjusted SFS, we ran forward simulation with mutations under multiple selective constraints, and we tested what ratio of neutral to selected mutations can lead to an age-adjusted SFS similar to that observed for retrotransposons in *B. distachyon*. Specifically, we investigated the conditions under which we observed a Δ frequency in the oldest age bin significantly smaller than Δ frequency in all other age bins. Our simulations revealed that the shape of the age-adjusted SFS of retrotransposons could only be reproduced if less than 10% of the mutations were neutrally evolving for most of the selective constraint investigated (*Figure 2—figure supplement 2* and *Supplementary file 1-table 1d*).

Finally, we used the results from our simulations to narrow down the selection strength affecting retrotransposons in *B. distachyon* by investigating the age of the oldest retrotransposons in our dataset. The main difference between the age-adjusted SFS of mutations evolving under weak and strong purifying selection is that the oldest mutations are much older in the simulation with weak purifying selection than in the simulation with strong purifying selection. This age difference arises because mutations under strong purifying selection are removed from the population more effectively and, therefore, cannot persist as long in the population. Examining the age of the last retrotransposon

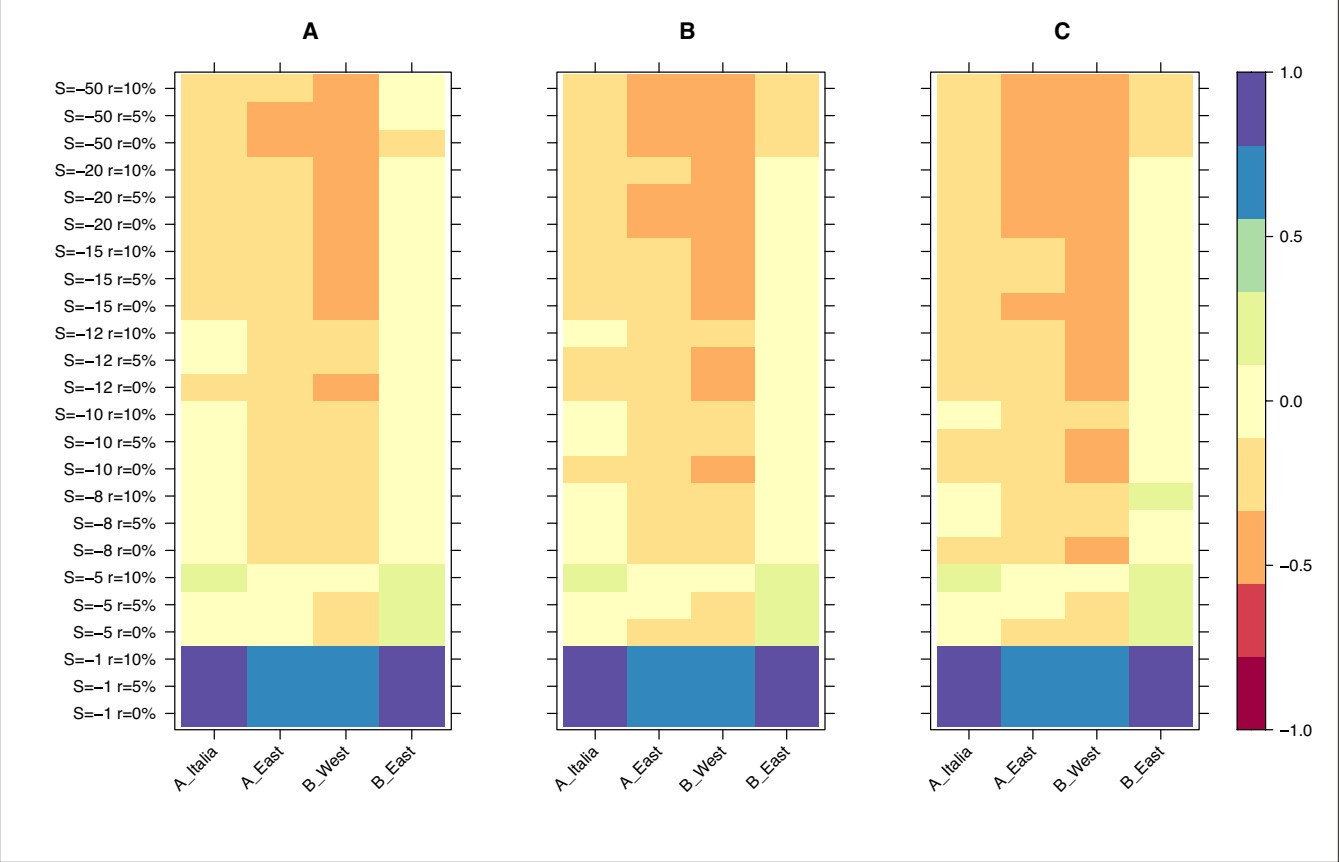

**Figure 3.** Relative age difference ((mutation age in simulations - observed mutation age)/maximum absolute age difference) between simulated and observed data in the last bin of the age-adjusted SFS. (**A**): 25% quantile; (**B**): 50% quantile; (**C**): 75% quantile. Relative age difference between simulated data assuming fully outcrossing individuals and observed data in the last bin of the age-adjusted SFS are shown in *Figure 3—figure supplement 1*.

The online version of this article includes the following figure supplement(s) for figure 3:

**Figure supplement 1.** Relative age difference ((mutation age in simulations - observed mutation age)/maximum absolute age difference) between simulated data assuming fully outcrossing individuals and observed data in the last bin of the age-adjusted SFS.

bins in the age-adjusted SFS revealed that the ages of the oldest retrotransposons were the most similar to the expected ages of the oldest mutations in our simulations, with a scaled selection coefficient (S) of –5 and –8 (*Figure 3*), indicating that retrotransposons in *B. distachyon* are under moderate purifying selection. In simulations with a nearly neutral selection coefficient (S = –1), the simulated mutations were much older than the oldest observed retrotransposons (*Figure 3*). Conversely, in simulations with a strong purifying selection coefficient (S < –10), they were much younger than the oldest observed retrotransposons (*Figure 3*).

## Discussion

*B. distachyon* is a widely used model species in evolutionary genomics, molecular ecology, developmental biology, and crop functional genomics (for review *Raissig and Woods, 2022*; *Hasterok et al., 2022*) with past and ongoing TE movements in its genome (*Lockton et al., 2008*). In this study, we used a diversity panel containing next-generation sequencing data from over 320 individuals sampled across the whole geographical range of *B. distachyon* to examine the role of TEs during evolution and adaptation. We investigated the frequency with which positive selection led to an increase in the frequency and fixation of TEs and quantified the strength of purifying selection on TE polymorphisms. Accounting for population structure and fluctuant transposition rates, we demonstrate that TEs are rarely part of the genetic makeup that was positively selected during environmental adaptation in *B. distachyon*. Furthermore, we show that the majority of TE polymorphisms found in the natural

population of this model species are under weak to moderate purifying selection, with only a small minority of TE polymorphisms evolving neutrally.

## Rare instances of positive selection on TEs

By combining complementary approaches, we were able to demonstrate that TEs are rarely the target of positive selection in *B. distachyon*. We first probed for footprints of positive selection on TE polymorphisms using the five genetic clades as focal populations. In conducting this analysis, we did not find TE polymorphisms to be at high frequencies or fixed at higher rates than expected, in regions of the genome presumably harboring selective sweeps in at least one of the genetic clades (high iHS regions). This suggested that TEs were rarely the target of positive selection, which we confirmed with a genome-wide scan for overly differentiated TE polymorphisms using $X^tX$ analysis. Indeed, this approach revealed that only a very small proportion of TE polymorphisms are more differentiated than expected under a neutral scenario.

Importantly, the $X^tX$ analysis also revealed that a non-negligible fraction of the TE polymorphisms is less differentiated than expected and are shared among genetic clusters. This could be the result of selection favoring the same TE polymorphisms in different accessions to adapt to similar environmental constraints across genetic clades. To test this scenario, we performed GEA with 32 environmental factors, and found only nine TE polymorphisms significantly associated with any of these, and representing a very small proportion (<0.01%) of all the TE polymorphisms we identified. Interestingly though, these nine TE polymorphisms were associated with environmental variables pertaining to precipitation, temperature and altitude, which are known to drive adaptation in *B. distachyon* (*Minadakis et al., 2023*). Some insertions were found within or in close proximity of genes, making these polymorphisms very good candidates for future functional validation.

Single TE insertions can have a drastic impact on phenotypic variation and be affected by positive selection (for review, see *Dubin et al., 2018*; *Bourgeois and Boissinot, 2019*; *Casacuberta and González, 2013*). For instance, TEs have increased in frequency through positive selection in humans (*Jiang et al., 2022*) or during range expansion in *Arabidopsis* (*Castanera et al., 2023*) and *D. melanogaster* (*Barrón et al., 2014*; *Niu et al., 2019*). Evidently, *B. distachyon* exhibits a different pattern, as causal mutations for adaptation in this grass species are rarely TEs. Only a few studies have thoroughly quantified the extent to which positive selection influences the evolution of TEs (*Castanera et al., 2023*; *Bourgeois et al., 2020*; *Charlesworth, 1991*; *Charlesworth and Charlesworth, 1983*; *Charlesworth, 1996*). But two of these drew similar conclusions to us, in the green anole *Anolis carolinensis* (*Charlesworth and Charlesworth, 1983*) and in the invasive species *Drosophila Suzukii* (*Charlesworth, 1996*). In addition, a large number of candidate genes for adaptation were identified with a similar approach focusing on SNPs (*Minadakis et al., 2023*), indicating that population structure or demographic events are not limiting factors for the methods we used. Altogether, these observations call for a closer investigation of which forces, for example purifying selection or neutral evolution, are important in shaping TE allele frequency in natural populations.

## Moderate purifying selection is the dominant force during TE evolution

Our results suggest that purifying selection is an important factor limiting the ability of TE polymorphisms to fix and increase their frequency in *B. distachyon*. This finding that purifying selection is the main force shaping the landscape of TE polymorphisms in *B. distachyon* is in line with similar observations made for example in maize (*Stitzer et al., 2023*), *Arabidopsis thaliana* (*Quadrana et al., 2016*; *Baduel et al., 2021*) and *Drosophila simulans* (*Langmüller et al., 2023*). Indeed, one of the significant explanatory variables in our ANCOVA models was the genetic clade, a proxy for the effective population size ($N_e$), which affects the efficiency with which selection can fix beneficial mutations and purge deleterious ones. In *B. distachyon*, the number of fixed TE polymorphisms per clade and the frequency of TE polymorphisms were negatively correlated with $N_e$, indicating that the accumulation of TEs is significantly lower in genetic clades with a larger $N_e$, potentially because of a greater efficacy of purifying selection.

It is widely accepted that most new TE insertions have a deleterious or no effect on the fitness of the host (*Niu et al., 2019*; *Bourgeois et al., 2020*; *Charlesworth, 1991*; *Charlesworth and Charlesworth, 1983*; *Boissinot et al., 2006*; *Blumenstiel et al., 2014*; *Rech et al., 2019*; *Mérel et al., 2021*; *Langmüller et al., 2023*). To properly quantify the effect of purifying selection on TE

evolution in *B. distachyon*, we used age-adjusted SFS analyses to evaluate the selective constraint experienced by TE polymorphisms while accounting for previously reported changes in their activity (*Lockton et al., 2008*). While this method can only be applied to retrotransposons (because the model does not allow back mutations), it provided a first clue on the importance of purifying selection on TE evolution and revealed that overall, retrotransposons evolved under purifying selection in all four derived genetic clades. Indeed, the Δ frequency was significantly smaller than 0, especially for older retrotransposons, meaning that old retrotransposons are less common than neutrally evolving SNPs at the same age. This further demonstrates that even after accounting for the different genetic clades and using a large sample size, retrotransposons evolve under purifying selection in *B. distachyon*. One caveat of the approach used in this study is that TE calling pipelines based on short-reads tend to have higher false positive and false negative call rates than SNP calling pipelines, which is also the case for the TEPID TE calling pipeline used here (*Baduel et al., 2021*; *Stuart et al., 2016*). A high false negative TE calling rate however might bias our TE frequency estimates toward lower frequencies, which could drive the observed patterns in the age-adjusted SFS. To assess if the false negative TE calling rate in our study substantially affected our results, we re-run the age-adjusted SFS on a subset of our dataset only including samples with a genome-wide mapping coverage of at least 20 x, as higher mapping coverages are expected to reduce the false negative call rate (*Stritt et al., 2018*; *Stuart et al., 2016*). Using the TE allele frequencies estimated based on this subset of our data to estimate Δ frequency revealed similar results of the age-adjusted SFS based on the whole dataset (*Figure 2—figure supplement 3*), indicating that our observation of retrotransposons evolving under purifying selection is not solely driven by a high false negative TE calling rate.

We also revealed that only a minority of retrotransposons evolved neutrally, as the observed shape of the Δ frequency curve could only be reproduced in our simulation if the proportion of neutrally evolving mutations in our focal mutations was below 10%. This estimate gives a first glimpse into the distribution of fitness effects of new TE insertions, a fundamental parameter in genetics that describes the way in which new TE insertions can contribute to evolution and adaptation (*Eyre-Walker et al., 2006*). Here, we show for the first time that new TE insertions have a less than 10% chance to insert into the genome of *B. distachyon* in a way that will allow them to evolve neutrally, advocating for a large potential of TEs to create, through their movement, new phenotypic variation on which selection can act on. PCAs based on TE polymorphisms allowed us to recover the population structure of *B. distachyon*, implying that demographic history and hence neutral processes may indeed partially explain the differences in the TE distribution we observed between genetic clades, as shown in *Arabidopsis thaliana* and *Arabidopsis lyrata* (*Bourgeois et al., 2020*; *Lockton and Gaut, 2010*), *Drosophila melanogaster* (*González et al., 2009*), humans (*Xue et al., 2018*) and the green anole (*Anolis carolinensis*; *Charlesworth and Charlesworth, 1983*). However, and in line with our simulations, the first two axes of the PCA explain less than 7% of the variance, indicating that neutrally evolving TEs contribute only mildly to overall TE diversity in our system.

Because TEs can cause phenotypic variation through new insertions (*Bhattacharyya et al., 1990*; *Hof et al., 2016*; *Feschotte, 2008*; *Qiu and Köhler, 2020*; *Slotkin and Martienssen, 2007*; *Hollister and Gaut, 2009*; *Xiao et al., 2008*; *Gordon et al., 2017*), it is not surprising that most new insertions interfere with the function of the genome, especially in a species with a small genome, such as *B. distachyon* (272 Mb) (*International Brachypodium Initiative, 2010*). The proportion of neutrally evolving TE polymorphisms is expected to be very small in genes, as insertions in genic regions are likely to result in loss-of-function (*Bhattacharyya et al., 1990*; *Hof et al., 2016*). Similarly, TE insertions in close proximity to genes are expected to be highly disruptive, as regulatory elements such as *cis*-regulatory elements are predominantly located in the proximity of genes. In *A. thaliana*, for instance, TEs located in the vicinity of genes (less than 2 kb) globally result in downregulation (*Wang et al., 2013*). Although only specific families alter gene expression in *B. distachyon* (*Wyler et al., 2020*), the observed Δ frequency for retrotransposon polymorphisms in genes and in their 1 kb surroundings matched our expectations. The fact that TE polymorphisms located more than 5 kb away from genes are also evolving under purifying selection was more surprising. That said, little is known about the distance between *cis*-regulatory sequences and genes in *B. distachyon*. In plants, TEs are believed to affect gene expression in trans through the production of small-interfering RNA (*McCue et al., 2013*; *McCue et al., 2012*; *McCue and Slotkin, 2012*; *Cho, 2018*; *Wyler et al., 2022*). Hence, the fact that only a small proportion of TEs can accumulate neutrally indicates that, in a gene-dense genome

such as that of *B. distachyon* (42.5% of the genome are genes; *Wyler et al., 2022*), TE insertions in any genomic compartment may result in some *cis-* or *trans*-regulatory effects visible to selection. Finally, we tested whether TE polymorphisms located more than 5 kb away from genes are evolving under purifying selection could be due to mapping or other artefacts by comparing the shape of the age-adjusted SFS of retrotransposons and SNPs more than 5 kb away from genes. However, the age-adjusted SFS of SNPs 5 kb away from genes differs from the one of retrotransposons (*Figure 2—figure supplement 4*), indicating that the shape of the age-adjusted SFS of retrotransposons more than 5 kb away from genes is not likely to be the result of artefacts in regions of the genome far away from genes.

To further ascertain the strength of purifying selection, we used forward simulation and showed that simulations assuming a moderately weak selection pressure (S = –5 or S = –8) against TE polymorphisms best fitted our observed data. In theory, no TE polymorphisms under strong purifying selection should be present in a natural population, as such mutations are expected to be quickly lost, especially in a predominantly selfing species where most loci are expected to be homozygous. Therefore, it is not surprising that TE polymorphisms which persist in *B. distachyon* are under weak to moderate selection, as also shown, for example, for the L1 retrotransposons in humans (*Stritt et al., 2018*) or the BS retrotransposon family in *Drosophila melanogaster* (*González et al., 2009*).

While some of the parameters we chose for our simulations, such as the dominance or selfing rate, can affect the efficiency of TE purging, it is unlikely that discrepancies in the true and assumed values for these parameters would have led to drastically different results. For example, we assumed codominance for all mutations, which might not hold true for each TE polymorphism. However, because of the high selfing rate observed in *B. distachyon* (*Stritt et al., 2022*), heterozygous loci are expected to be rare, and dominance is unlikely to have a strong impact on our observations. Similarly, with a higher selfing rate, deleterious TE polymorphisms should be removed more efficiently by purifying selection. To check whether a lower selfing rate could allow a higher proportion of TE polymorphisms to evolve neutrally, we reran the simulations assuming fully outcrossing individuals. This also resulted in simulation with weak to moderate selection strength on TE polymorphisms best fitting the observed data, further strengthening our results (*Figure 3—figure supplement 1*).

While the analyses of positive selection and GEA were based on both DNA-transposons and retrotransposons, we only used retrotransposons to assess the strength of selection on TE polymorphisms, as the age-adjusted SFS was developed with the assumption of no back mutations (*Horvath et al., 2022*). Yet, DNA-transposons do not solely transpose through cut and paste mechanisms as they would otherwise not be so abundant in Eukaryotic genomes. DNA-transposons can also create extra copies of themselves by transposing during chromosome replication or repair from a position that has already been replicated, or repaired (*Wicker et al., 2007*). We therefore repeated the age-adjusted SFS analyses using DNA-transposons to evaluate whether DNA-transposons were affected by similar selective constraints. The folded SFS of DNA-transposons and retrotransposons display similar shifts toward high proportions of rare alleles and Δ frequency deviations from 0 in the age-adjusted SFS of DNA-transposons and retrotransposons are comparable. Hence, we argue that the conclusion drawn for retrotransposons also holds for DNA-transposons, and that purifying selection affect TEs broadly. To further examine our conclusion on purifying selection, we investigated the selective regime affecting different retrotransposons and DNA-transposons superfamilies. Thereby, we generated age-adjusted SFS for the four most common TE superfamilies Copia, Ty3 (also known under the name Gypsy, but we will avoid using this name because of its problematic nature see *Wei et al., 2022*), Helitron and MITE and found similar deviations of the Δ frequency from 0 in the four investigated TE superfamilies (*Figure 2—figure supplements 5–8*). These results indicate that our conclusion on the broad effect of purifying selection is not driven by a single TE superfamily but is at least common among the four most numerous TE superfamilies.

## Conclusion

Adaptation to different environmental conditions is a complex process that involves various mutation types. Here, we show that the vast majority of TE polymorphisms are under purifying selection in the small genome of *B. distachyon*. Conversely, only a very small proportion of TEs seem to have contributed to adaptation. The observed lack of neutrally evolving TE polymorphisms in *B. distachyon* advocates for a large potential of TE polymorphisms to contribute to the genetic diversity and phenotypic

variation on which selection can act and highlights the need to consider TE polymorphisms during evolutionary studies. Finally, our work shows that the ability of TEs to cause phenotypic variation does not necessarily translate into being favored during evolution and adaptation over other mutations with more subtle effects, such as SNPs.

## Materials and methods
### Whole-genome resequencing data
In this study, we analyzed a total of 326 publicly available whole-genome sequencing data from *Brachypodium distachyon* accessions sampled around the Mediterranean Basin (*Figure 1A*; *Supplementary file 1-table 1e*). Our *B. distachyon* dataset consisted of 47 samples published by *Gordon et al., 2017*, 57 samples published by *Skalska et al., 2020*, 65 samples published by *Gordon et al., 2020*, 86 samples published by *Stritt et al., 2022* and 71 samples published by *Minadakis et al., 2023*, covering all five genetic clades previously described in this species (*Stritt et al., 2022*; *Minadakis et al., 2023*). Each sample was assigned to a genetic clade based on previously published results (*Minadakis et al., 2023*).

### Data processing
Raw reads were trimmed using Trimmomatic 0.36 (*Bolger et al., 2014*) and mapped to the *B. distachyon* reference genome version 3.0 (*International Brachypodium Initiative, 2010*) using bowtie2 (*Langmead and Salzberg, 2012*) and yaha (*Faust and Hall, 2012*), and TE polymorphisms were identified using the TEPID pipeline (*Stuart et al., 2016*) and the recently updated TE annotation by *Stritt et al., 2020* and *Wyler et al., 2022*. TE polymorphisms include both TE insertion polymorphisms (TIPs; insertions absent from the reference genome but present in at least one natural accession) and TE absence polymorphisms (TAPs; insertions present in the reference genome but absent from at least one natural accession). The class, superfamily and family of each TE call were assigned based on the TEPID results and the TE annotation from the reference genome. TIPs that were less than 100 base pairs (bp) apart in different samples and assigned to the same TE family were merged.

SNPs were called using GATK v.4.0.2.1 (*McKenna et al., 2010*) using HaplotypeCaller (*Poplin et al., 2018*) following *Minadakis et al., 2023*. The SNP calls were hard filtered using the following conditions: QD <5.0; FS >20.0; SOR >3.0; MQ <50.0; MQRankSum <2.5; MQRankSum >–2.5; ReadPosRankSum <2.0; ReadPosRankSum >–2.0. Because *B. distachyon* displays a high selfing rate (*Stritt et al., 2022*), most genetic variants are expected to be homozygous within an individual. Hence, all TE calls were treated as homozygous, and heterozygous SNP calls were removed from our dataset to reduce false variant calls. Additionally, all sites with multiallelic TE and SNP calls were removed. SNPs were classified as synonymous, non-synonymous and of high fitness effect using SnpEff (*Cingolani et al., 2012*). SNPs and TE polymorphisms were merged into a single vcf file using custom scripts provided in GitHub (copy archived at *Roberthorv, 2024*).

To estimate the age of each SNP and TE polymorphism, the SNPs and TEs found in the A_East, A_Italia, B_East and B_West clades were polarized using the C clade, which was identified as the most ancestral *B. distachyon* clade (*Stritt et al., 2022*) and used as the outgroup throughout this study. An estimate for the time of origin of all SNPs and TE polymorphisms was calculated with GEVA, a nonparametric approach that relies on pairwise differences in identity by descent (IBD) regions around the focal mutation to estimate the time of origin (*Albers and McVean, 2020*). GEVA was run separately for each clade using the genetic map produced by *Huo et al., 2011* and a mutation rate of $7\times10^{-9}$ substitutions/generation. The theoretical prediction of the correlation between allele age and allele frequency of neutrally evolving mutations based on $N_e$ (*Kimura and Ohta, 1973*) was compared to the observed correlation between allele age and frequency of synonymous SNPs to check the sanity of the age estimates.

The observed SNP and TE diversity was first examined using a principal component analysis (PCA), and correlations between TE diversity and genetic clades were tested with a mantel test using the ade4 package version 1.7–22 (*Dray and Dufour, 2007*) in R version 4.1.2 (*R Development Core Team, 2021*). The folded site frequency spectrum (SFS) was computed for TE polymorphisms and SNPs using the minor allele frequency in R version 4.1.2 (*R Development Core Team, 2021*). Finally,

the map of the geographical distribution of the used accessions was done in R using the rnaturalearth package 0.3.3 (*Massicotte and South, 2023*).

Mapping coverage is known to influence false discovery rate (*Stritt et al., 2018*; *Stuart et al., 2016*). To investigate the impact of false positive and false negative TE calls on our results, we down sampled the TE dataset to only include TEs that have been called in samples that had at least an average mapping coverage of 20 x. The allele frequencies of TEs present in our high coverage dataset was recalculated only considering samples with at least an average mapping coverage of 20 x. This second TE dataset was then used to check if using a dataset with a higher mapping coverage and presumably a lower false TE calling rate impacted our results.

## Analyses of positive selection

Regions of the genome affected by positive selection were identified using the integrated haplotype score (iHS), a measure of the amount of extended haplotype homozygosity along the ancestral allele relative to the derived allele for a given polymorphic site (*Sabeti et al., 2002*). iHS was calculated using the SNP dataset, and regions displaying longer haplotypes and hence high iHS were identified in R using the rehh package (*Gautier et al., 2017*; *Gautier and Vitalis, 2012*). The threshold to distinguish between regions of high iHS and other regions was selected such that less than 5% of the *B. distachyon* genome was classified as high iHS regions in each clade (*Supplementary file 1-table 1f*). Candidate regions under positive selection were defined as all regions that were found to have high iHS in each clade separately.

A first ANCOVA was used to model the number of fixed TE polymorphisms in each clade found in the candidate region under positive selection based on the following genetic features: total number of TEs, TE superfamily, TE age (split into three categories: young: age <10,000 generations; intermediate: age between 10,000 generations and 60,000 generations; old: age >60,000 generations), clade, genomic region (a unique ID for each candidate region under positive selection) and iHS classification of the regions in each clade (high or average). A second ANCOVA was used to model the allele frequency of TE polymorphisms found in the candidate region under positive selection based on the following genetic features: TE superfamily, TE age, clade, genomic region and iHS classification of the regions in each clade. The TE superfamily was included to account for different evolutionary behaviors of TEs from different superfamilies. Age accounted for differences in the fixation rate and frequency distribution between young and old TEs. The clade was included to account for clade-specific differences such as differences in $N_e$. Finally, a unique ID for each candidate region under positive selection was included to account for region-specific differences such as differences in the recombination rate and GC content. In the end, regions that were found to have a high iHS in some clades were compared to the same regions in the other clades. All ANCOVAs were run in R using the car package (*Fox and Weisberg, 2019*).

The standardized allele frequency of a mutation across populations ($X^tX$) values (*Günther and Coop, 2013*) were computed for the combined TE and SNP dataset using Baypass version 2.3 (*Olazcuaga et al., 2020*; *Gautier, 2015*). The $X^tX$ values were used to identify over- and under differentiated TE polymorphisms between clades. A pseudo-observed dataset (POD) of 100,000 SNPs was simulated under the demographic model inferred from the covariance matrix of the SNP dataset. The POD was then used to determine the 97.5% (over-differentiated polymorphisms) and 2.5% (under differentiated polymorphisms) quantiles.

## Genome-environment association analyses

We identified TE polymorphisms significantly associated with environmental factors using genome-environment association analyses (GEA) following *Minadakis et al., 2023*. GEAs were run with GEMMA 0.98.5 (*Zhou and Stephens, 2012*) using the combined TE and SNP vcf file against the following 32 environmental factors extracted by *Minadakis et al., 2023*: altitude, aridity from March to June, aridity from November to February, annual mean temperature, mean temperature of warmest quarter, mean temperature of coldest quarter, annual precipitation, precipitation of wettest month, precipitation of driest month, precipitation seasonality, precipitation of wettest quarter, precipitation of driest quarter, precipitation of warmest quarter, precipitation of coldest quarter, mean diurnal Range, isothermality, temperature seasonality, maximum temperature of warmest month, minimum temperature of coldest month, temperature annual range, mean temperature of wettest quarter, mean temperature of driest

quarter, precipitation from March to June, precipitation from November to February, solar radiation from March to June, solar radiation from November to February, mean temperature between March and June, mean temperature between November and February, maximum temperature between March and June, maximum temperature between November and February, minimum temperature between March and June and minimum temperature between November and February. We applied a False Discovery Rate (FDR, *Benjamini and Hochberg, 1995*) threshold of 5% to control for false positive rates.

## Age-adjusted frequency spectra and analyses of purifying selection

Footprints of purifying selection on TE polymorphisms were first evaluated using folded SFS. An age-adjusted site frequency spectrum (age-adjusted SFS) approach was used to further investigate the impact of purifying selection on retrotransposons while accounting for nonconstant transposition rates. Briefly, the age-adjusted SFS is a summary statistic that describes the difference between the average frequency of TEs at a specific age and the average frequency of neutral sites of the same age (*Horvath et al., 2022*). Therefore, the TE dataset was sorted by age and split into equally large bins with respect to the number of observations in each age bin. Neutral sites were then randomly down-sampled to match the number of observations in the TE dataset and its age distribution (*Horvath et al., 2022*).

The difference between the average TE and neutral site frequency, or Δ frequency, was computed for each age bin (*Horvath et al., 2022*). This method allows for an unbiased comparison between the allele frequencies of TEs and neutral sites, and is robust to transposition rate changes and demographic changes (*Horvath et al., 2022*). However, the theory behind this method was developed assuming no back mutations and is therefore best suited for retrotransposons, as DNA-transposons can exit an insertion site (*Horvath et al., 2022*). We used the synonymous SNPs identified with SnpEff as the neutrally evolving sites. However, because estimating the population wide frequency of TEs is more challenging than estimating SNP frequencies, putative biases in frequency estimates need to be assessed before performing age-adjusted SFS analyses. To do so, the SNP dataset was resampled so that the SNP dataset used in the age-adjusted SFS had a frequency distribution that matched the observed TE frequency distribution. The age-adjusted SFS of retrotransposons was contrasted against the age-adjusted SFS of non-synonymous, as well as against high fitness effect SNPs. Therefore, 10,000 non-synonymous and high fitness effect SNPs were randomly selected for each clade to reach approximately the same number of retrotransposon polymorphisms, non-synonymous and high fitness effect SNPs for final comparisons. To estimate the variation in Δ frequency estimates, all age-adjusted SFS were computed 100 times. All Wilcoxon tests and Bonferroni p value corrections were done in R version 4.1.2 (*R Development Core Team, 2021*).

## Forward simulation

We used SLiM 4.0.1 (*Haller and Messer, 2019a*; *Haller and Messer, 2019b*) to run forward simulations and assess the proportion of neutrally evolving retrotransposons and the average selection strength affecting them. The simulations were designed to reflect the population size and demographic history of *B. distachyon*. The simulated genomic fragment was 1 megabase (Mb) long and included neutral (synonymous) mutations as well as focal mutations that evolved under different selective constraints. The focal mutations were a mix of neutrally evolving mutations and mutations evolving under a constant selection pressure. Therefore, the ratio (r) of focal mutations that evolved neutrally was either 0%, 5%, 10%, 25% or 50%. The scaled selection coefficient (S, defined as $N_e s$, with $s$ the strength of selection and $N_e$ the effective population size) affecting the remaining focal mutations was set at the beginning of the simulation to be either −1,−5, −8,−10, −12,−15, −20 or −50 to cover effectively neutral (0>S ≥ −1), intermediate (−1>S ≥ −10) ,and strongly deleterious (−10>S) selective constraints. The selfing rate was set to 70%, as *B. distachyon* is a highly selfing species with occasional outcrossing (*Stritt et al., 2022*; *Minadakis et al., 2023*). In addition, a high recombination rate was chosen to minimize the effects of linked selection in the small genomic fragment simulated. Simulations for each combination of these two parameters were run 20 times to assess the variation in the resulting age-adjusted SFS. The shape of the resulting age-adjusted SFS was used to narrow down the ratio of neutrally evolving TE polymorphisms. Similarly, the age distribution of the mutations

in the oldest bin of the age-adjusted SFS was used to narrow down the strength of selection affecting TE polymorphisms.

## Acknowledgements

We thank Jeffrey Ross-Ibarra and Mitra Menon as well as Fabrizio Menardo, Michael Thieme, Wenbo Xu, Jigisha, Lars Kaderli and Serafin Schefer for all the discussions and their comments on this project. We thank Emmanuelle Botté (https://manuscribe.com.au) for professional editing of the manuscript. Funding: This project was funded by the Swiss National Science Foundation (project 31003A_182785) and the Research Priority Program Evolution in Action from the University of Zürich. Data analyzed in this paper were generated in collaboration with the Genetic Diversity Center (GDC), ETH Zürich.

## Additional information

### Funding

| Funder | Grant reference number | Author |
| --- | --- | --- |
| University of Zurich Foundation | Research Priority Program Evolution in Action | Anne C Roulin |
| Swiss National Science Foundation | 31003A_182785 | Anne C Roulin |

The funders had no role in study design, data collection and interpretation, or the decision to submit the work for publication.

### Author contributions

Robert Horvath, Conceptualization, Data curation, Formal analysis, Project administration, Validation, Visualization, Writing – original draft, Writing – review and editing; Nikolaos Minadakis, Data curation, Formal analysis; Yann Bourgeois, Formal analysis, Validation; Anne C Roulin, Conceptualization, Data curation, Funding acquisition, Methodology, Project administration, Supervision, Writing – original draft, Writing – review and editing

### Author ORCIDs

Robert Horvath ⬛ https://orcid.org/0000-0002-3221-8835
Yann Bourgeois ⬛ https://orcid.org/0000-0002-1809-387X
Anne C Roulin ⬛ https://orcid.org/0000-0002-6668-3321

Reviewer #1 (Public Review): https://doi.org/10.7554/eLife.93284.3.sa1
Reviewer #2 (Public Review): https://doi.org/10.7554/eLife.93284.3.sa2
Author response https://doi.org/10.7554/eLife.93284.3.sa3

## Additional files

### Supplementary files

• Supplementary file 1. Supplementary tables. Table 1a ANCOVA predicting the number of fixed TE polymorphisms per clade in candidate regions under positive selection in accessions with at least 20 x coverage. Table 1b ANCOVA predicting the allele frequency of TE polymorphisms per clade in candidate regions under positive selection in accessions with at least 20 x coverage. Table 1c List of TEs significantly associated with at least one environmental factor in the GWAS. The "Gene in the proximity" columns include information on the genes in the proximity of the TE insertion (less than 2 kb up and downstream). The last five columns indicate the frequency of the TE in each clade. Table 1d Difference in delta frequency between the oldest and second oldest age bin in the different simulations. Table 1e List of published samples used in this study. Because the reference accession was sequenced in multiple study and some samples were identified as outliers (indicating a wrong species classification or hybrid individuals) using PCA analyses on SNP and TE calls, only the samples listed below from each previous study were used. Table 1f List of thresholds used and percentage of

the genome classified as high iHS regions in the four derived clades.

• MDAR checklist

## Data availability

The datasets supporting the conclusions of this article are publicly available on the European Nucleotide Archive and National Center for Biotechnology Information, and the archive numbers of the accessions used are listed in the *Supplementary file 1-table 1e*. The scripts generated are available on GitHub, copy archived at *Roberthorv, 2024*.

The following previously published datasets were used:

| Author(s) | Year | Dataset title | Dataset URL | Database and Identifier |
|---|---|---|---|---|
| Skalska A, Stritt C, Wyler M, Williams HW, Vickers M, Han J, Tuna M, Tuna GS, Susek K, Swain M, Wóycicki RK, Chaudhary S, Corke F, Doonan JH, Roulin AC, Hasterok R, Mur LAJ | 2020 | Genetic and epigenetic variation in a model grass Brachypodium distachyon | https://www.ncbi.nlm.nih.gov/bioproject/PRJNA605320/ | NCBI BioProject, PRJNA605320 |
| Gordon SP, Contreras-Moreira B, Levy JJ, Djamei A, Czedik-Eysenberg A, Tartaglio VS, Session A, Martin J, Cartwright A, Katz A, Singan VR, Goltsman E, Barry K, Dinh-Thi VH, Chalhoub B, Diaz-Perez A, Sancho R, Lusinska J, Wolny E, Nibau C, Doonan JH, Jenkins CPJ, Hazen SP, Lee SJ, Shu S, Goodstein D, Rokhsar D, Schmutz J, Hasterok R, Catalan P, Vogel JP, Mur LAJ | 2020 | Brachypodium distachyon strain:Koz-2 | cultivar:Koz-2 (stiff brome) | https://www.ncbi.nlm.nih.gov/bioproject/PRJNA333752 | NCBI BioProject, PRJNA333752 |
| Stritt C, Gimmi EL, Wyler M, Bakali AH, Skalska A, Hasterok R, Pecchioni N, Roulin AC, Mur LAJ | 2022 | Whole genome sequencing of natural populations of the grass Brachypodium distachyon, collected in Italy and France | https://www.ebi.ac.uk/ena/browser/view/PRJEB40344 | EBI European Nucleotide Archive, PRJEB40344 |
| Nikolaos M, Hefin W, Horvath R, Caković D, Stritt C, Thieme M, Bourgeois Y, Roulin AC | 2023 | Brachypodium distachyon 332 | https://www.ebi.ac.uk/ena/browser/view/PRJEB61986 | EBI European Nucleotide Archive, PRJEB61986 |

*Continued on next page*

*Continued*

| Author(s) | Year | Dataset title | Dataset URL | Database and Identifier |
|---|---|---|---|---|
| Gordon SP, Contreras-Moreira B, Levy JJ, Djamei A, Czedik-Eysenberg A, Tartaglio VS, Session A, Martin J, Cartwright A, Katz A, Singan VR, Goltsman E, Barry K, Dinh-Thi VH, Chalhoub B, Diaz-Perez A, Sancho R, Lusinska J, Wolny E, Nibau C, Doonan JH, Jenkins CPJ, Hazen SP, Lee SJ, Shu S, Goodstein D, Rokhsar D, Schmutz J, Hasterok R, Catalan P, Vogel JP, Mur LAJ | 2014 | Brachypodium distachyon strain:Arn1 (stiff brome) | https://www.ncbi.nlm.nih.gov/bioproject/PRJNA249894 | NCBI BioProject, PRJNA249894 |
| Gordon SP, Contreras-Moreira B, Levy JJ, Djamei A, Czedik-Eysenberg A, Tartaglio VS, Session A, Martin J, Cartwright A, Katz A, Singan VR, Goltsman E, Barry K, Dinh-Thi VH, Chalhoub B, Diaz-Perez A, Sancho R, Lusinska J, Wolny E, Nibau C, Doonan JH, Jenkins CPJ, Hazen SP, Lee SJ, Shu S, Goodstein D, Rokhsar D, Schmutz J, Hasterok R, Catalan P, Vogel JP, Mur LAJ | 2014 | Brachypodium distachyon Luc1 (stiff brome) | https://www.ncbi.nlm.nih.gov/bioproject/PRJNA249901 | NCBI BioProject, PRJNA249901 |
| Gordon SP, Contreras-Moreira B, Levy JJ, Djamei A, Czedik-Eysenberg A, Tartaglio VS, Session A, Martin J, Cartwright A, Katz A, Singan VR, Goltsman E, Barry K, Dinh-Thi VH, Chalhoub B, Diaz-Perez A, Sancho R, Lusinska J, Wolny E, Nibau C, Doonan JH, Jenkins CPJ, Hazen SP, Lee SJ, Shu S, Goodstein D, Rokhsar D, Schmutz J, Hasterok R, Catalan P, Vogel JP, Mur LAJ | 2014 | Brachypodium distachyon Per1 (stiff brome) | https://www.ncbi.nlm.nih.gov/bioproject/PRJNA249907 | NCBI BioProject, PRJNA249907 |

*Continued*

| Author(s) | Year | Dataset title | Dataset URL | Database and Identifier |
|---|---|---|---|---|
| Gordon SP, Contreras-Moreira B, Levy JJ, Djamei A, Czedik-Eysenberg A, Tartaglio VS, Session A, Martin J, Cartwright A, Katz A, Singan VR, Goltsman E, Barry K, Dinh-Thi VH, Chalhoub B, Diaz-Perez A, Sancho R, Lusinska J, Wolny E, Nibau C, Doonan JH, Jenkins CPJ, Hazen SP, Lee SJ, Shu S, Goodstein D, Rokhsar D, Schmutz J, Hasterok R, Catalan P, Vogel JP, Mur LAJ | 2024 | Brachypodium distachyon strain:Bd21-3 (stiff brome) | https://www.ncbi.nlm.nih.gov/bioproject/PRJNA250376 | NCBI BioProject, PRJNA250376 |
| Gordon SP, Contreras-Moreira B, Levy JJ, Djamei A, Czedik-Eysenberg A, Tartaglio VS, Session A, Martin J, Cartwright A, Katz A, Singan VR, Goltsman E, Barry K, Dinh-Thi VH, Chalhoub B, Diaz-Perez A, Sancho R, Lusinska J, Wolny E, Nibau C, Doonan JH, Jenkins CPJ, Hazen SP, Lee SJ, Shu S, Goodstein D, Rokhsar D, Schmutz J, Hasterok R, Catalan P, Vogel JP, Mur LAJ | 2014 | Brachypodium distachyon strain:ABR8 (stiff brome) | https://www.ncbi.nlm.nih.gov/bioproject/PRJNA258992 | NCBI BioProject, PRJNA258992 |
| Gordon SP, Contreras-Moreira B, Levy JJ, Djamei A, Czedik-Eysenberg A, Tartaglio VS, Session A, Martin J, Cartwright A, Katz A, Singan VR, Goltsman E, Barry K, Dinh-Thi VH, Chalhoub B, Diaz-Perez A, Sancho R, Lusinska J, Wolny E, Nibau C, Doonan JH, Jenkins CPJ, Hazen SP, Lee SJ, Shu S, Goodstein D, Rokhsar D, Schmutz J, Hasterok R, Catalan P, Vogel JP, Mur LAJ | 2010 | Brachypodium distachyon strain:Bd21 (stiff brome) | https://www.ncbi.nlm.nih.gov/bioproject/PRJNA32607 | NCBI BioProject, PRJNA32607 |

*Continued on next page*

*Continued*

| Author(s) | Year | Dataset title | Dataset URL | Database and Identifier |
|---|---|---|---|---|
| Gordon SP, Contreras-Moreira B, Levy JJ, Djamei A, Czedik-Eysenberg A, Tartaglio VS, Session A, Martin J, Cartwright A, Katz A, Singan VR, Goltsman E, Barry K, Dinh-Thi VH, Chalhoub B, Diaz-Perez A, Sancho R, Lusinska J, Wolny E, Nibau C, Doonan JH, Jenkins CPJ, Hazen SP, Lee SJ, Shu S, Goodstein D, Rokhsar D, Schmutz J, Hasterok R, Catalan P, Vogel JP, Mur LAJ | 2016 | Brachypodium distachyon cultivar:ABR3 (stiff brome) | https://www.ncbi.nlm.nih.gov/bioproject/PRJNA337130 | NCBI BioProject, PRJNA337130 |
| Gordon SP, Contreras-Moreira B, Levy JJ, Djamei A, Czedik-Eysenberg A, Tartaglio VS, Session A, Martin J, Cartwright A, Katz A, Singan VR, Goltsman E, Barry K, Dinh-Thi VH, Chalhoub B, Diaz-Perez A, Sancho R, Lusinska J, Wolny E, Nibau C, Doonan JH, Jenkins CPJ, Hazen SP, Lee SJ, Shu S, Goodstein D, Rokhsar D, Schmutz J, Hasterok R, Catalan P, Vogel JP, Mur LAJ | 2016 | Brachypodium distachyon cultivar:ABR4 (stiff brome) | https://www.ncbi.nlm.nih.gov/bioproject/PRJNA337131 | NCBI BioProject, PRJNA337131 |
| Gordon SP, Contreras-Moreira B, Levy JJ, Djamei A, Czedik-Eysenberg A, Tartaglio VS, Session A, Martin J, Cartwright A, Katz A, Singan VR, Goltsman E, Barry K, Dinh-Thi VH, Chalhoub B, Diaz-Perez A, Sancho R, Lusinska J, Wolny E, Nibau C, Doonan JH, Jenkins CPJ, Hazen SP, Lee SJ, Shu S, Goodstein D, Rokhsar D, Schmutz J, Hasterok R, Catalan P, Vogel JP, Mur LAJ | 2016 | Brachypodium distachyon cultivar:Uni2 (stiff brome) | https://www.ncbi.nlm.nih.gov/bioproject/PRJNA337138 | NCBI BioProject, PRJNA337138 |

*Continued*

| Author(s) | Year | Dataset title | Dataset URL | Database and Identifier |
|---|---|---|---|---|
| Gordon SP, Contreras-Moreira B, Levy JJ, Djamei A, Czedik-Eysenberg A, Tartaglio VS, Session A, Martin J, Cartwright A, Katz A, Singan VR, Goltsman E, Barry K, Dinh-Thi VH, Chalhoub B, Diaz-Perez A, Sancho R, Lusinska J, Wolny E, Nibau C, Doonan JH, Jenkins CPJ, Hazen SP, Lee SJ, Shu S, Goodstein D, Rokhsar D, Schmutz J, Hasterok R, Catalan P, Vogel JP, Mur LAJ | 2016 | Brachypodium distachyon cultivar:ABR7 (stiff brome) | https://www.ncbi.nlm.nih.gov/bioproject/PRJNA337145 | NCBI BioProject, PRJNA337145 |
| Gordon SP, Contreras-Moreira B, Levy JJ, Djamei A, Czedik-Eysenberg A, Tartaglio VS, Session A, Martin J, Cartwright A, Katz A, Singan VR, Goltsman E, Barry K, Dinh-Thi VH, Chalhoub B, Diaz-Perez A, Sancho R, Lusinska J, Wolny E, Nibau C, Doonan JH, Jenkins CPJ, Hazen SP, Lee SJ, Shu S, Goodstein D, Rokhsar D, Schmutz J, Hasterok R, Catalan P, Vogel JP, Mur LAJ | 2016 | Brachypodium distachyon cultivar:Adi-2 (stiff brome) | https://www.ncbi.nlm.nih.gov/bioproject/PRJNA337146 | NCBI BioProject, PRJNA337146 |
| Gordon SP, Contreras-Moreira B, Levy JJ, Djamei A, Czedik-Eysenberg A, Tartaglio VS, Session A, Martin J, Cartwright A, Katz A, Singan VR, Goltsman E, Barry K, Dinh-Thi VH, Chalhoub B, Diaz-Perez A, Sancho R, Lusinska J, Wolny E, Nibau C, Doonan JH, Jenkins CPJ, Hazen SP, Lee SJ, Shu S, Goodstein D, Rokhsar D, Schmutz J, Hasterok R, Catalan P, Vogel JP, Mur LAJ | 2016 | Brachypodium distachyon cultivar:Bis-1 (stiff brome) | https://www.ncbi.nlm.nih.gov/bioproject/PRJNA337153 | NCBI BioProject, PRJNA337153 |

*Continued on next page*

*Continued*

| Author(s) | Year | Dataset title | Dataset URL | Database and Identifier |
|---|---|---|---|---|
| Gordon SP, Contreras-Moreira B, Levy JJ, Djamei A, Czedik-Eysenberg A, Tartaglio VS, Session A, Martin J, Cartwright A, Katz A, Singan VR, Goltsman E, Barry K, Dinh-Thi VH, Chalhoub B, Diaz-Perez A, Sancho R, Lusinska J, Wolny E, Nibau C, Doonan JH, Jenkins CPJ, Hazen SP, Lee SJ, Shu S, Goodstein D, Rokhsar D, Schmutz J, Hasterok R, Catalan P, Vogel JP, Mur LAJ | 2016 | Brachypodium distachyon cultivar:Adi-12 (stiff brome) | https://www.ncbi.nlm.nih.gov/bioproject/PRJNA337348 | NCBI BioProject, PRJNA337348 |
| Gordon SP, Contreras-Moreira B, Levy JJ, Djamei A, Czedik-Eysenberg A, Tartaglio VS, Session A, Martin J, Cartwright A, Katz A, Singan VR, Goltsman E, Barry K, Dinh-Thi VH, Chalhoub B, Diaz-Perez A, Sancho R, Lusinska J, Wolny E, Nibau C, Doonan JH, Jenkins CPJ, Hazen SP, Lee SJ, Shu S, Goodstein D, Rokhsar D, Schmutz J, Hasterok R, Catalan P, Vogel JP, Mur LAJ | 2016 | Brachypodium distachyon cultivar:Adi-12 (stiff brome) | https://www.ncbi.nlm.nih.gov/bioproject/PRJNA337349 | NCBI BioProject, PRJNA337349 |
| Gordon SP, Contreras-Moreira B, Levy JJ, Djamei A, Czedik-Eysenberg A, Tartaglio VS, Session A, Martin J, Cartwright A, Katz A, Singan VR, Goltsman E, Barry K, Dinh-Thi VH, Chalhoub B, Diaz-Perez A, Sancho R, Lusinska J, Wolny E, Nibau C, Doonan JH, Jenkins CPJ, Hazen SP, Lee SJ, Shu S, Goodstein D, Rokhsar D, Schmutz J, Hasterok R, Catalan P, Vogel JP, Mur LAJ | 2016 | Brachypodium distachyon cultivar:Sig1 (stiff brome) | https://www.ncbi.nlm.nih.gov/bioproject/PRJNA337350 | NCBI BioProject, PRJNA337350 |

*Continued*

| Author(s) | Year | Dataset title | Dataset URL | Database and Identifier |
|---|---|---|---|---|
| Gordon SP, Contreras-Moreira B, Levy JJ, Djamei A, Czedik-Eysenberg A, Tartaglio VS, Session A, Martin J, Cartwright A, Katz A, Singan VR, Goltsman E, Barry K, Dinh-Thi VH, Chalhoub B, Diaz-Perez A, Sancho R, Lusinska J, Wolny E, Nibau C, Doonan JH, Jenkins CPJ, Hazen SP, Lee SJ, Shu S, Goodstein D, Rokhsar D, Schmutz J, Hasterok R, Catalan P, Vogel JP, Mur LAJ | 2016 | Brachypodium distachyon Jer1 (stiff brome) | https://www.ncbi.nlm.nih.gov/bioproject/PRJNA337354 | NCBI BioProject, PRJNA337354 |
| Gordon SP, Contreras-Moreira B, Levy JJ, Djamei A, Czedik-Eysenberg A, Tartaglio VS, Session A, Martin J, Cartwright A, Katz A, Singan VR, Goltsman E, Barry K, Dinh-Thi VH, Chalhoub B, Diaz-Perez A, Sancho R, Lusinska J, Wolny E, Nibau C, Doonan JH, Jenkins CPJ, Hazen SP, Lee SJ, Shu S, Goodstein D, Rokhsar D, Schmutz J, Hasterok R, Catalan P, Vogel JP, Mur LAJ | 2016 | Brachypodium distachyon cultivar:Bd18-1 (stiff brome) | https://www.ncbi.nlm.nih.gov/bioproject/PRJNA337356 | NCBI BioProject, PRJNA337356 |
| Gordon SP, Contreras-Moreira B, Levy JJ, Djamei A, Czedik-Eysenberg A, Tartaglio VS, Session A, Martin J, Cartwright A, Katz A, Singan VR, Goltsman E, Barry K, Dinh-Thi VH, Chalhoub B, Diaz-Perez A, Sancho R, Lusinska J, Wolny E, Nibau C, Doonan JH, Jenkins CPJ, Hazen SP, Lee SJ, Shu S, Goodstein D, Rokhsar D, Schmutz J, Hasterok R, Catalan P, Vogel JP, Mur LAJ | 2016 | Brachypodium distachyon cultivar:Bd2-3 (stiff brome) | https://www.ncbi.nlm.nih.gov/bioproject/PRJNA337363 | NCBI BioProject, PRJNA337363 |

*Continued*

| Author(s) | Year | Dataset title | Dataset URL | Database and Identifier |
|---|---|---|---|---|
| Gordon SP, Contreras-Moreira B, Levy JJ, Djamei A, Czedik-Eysenberg A, Tartaglio VS, Session A, Martin J, Cartwright A, Katz A, Singan VR, Goltsman E, Barry K, Dinh-Thi VH, Chalhoub B, Diaz-Perez A, Sancho R, Lusinska J, Wolny E, Nibau C, Doonan JH, Jenkins CPJ, Hazen SP, Lee SJ, Shu S, Goodstein D, Rokhsar D, Schmutz J, Hasterok R, Catalan P, Vogel JP, Mur LAJ | 2016 | Brachypodium distachyon cultivar:Bd21Control (stiff brome) | https://www.ncbi.nlm.nih.gov/bioproject/PRJNA337420 | NCBI BioProject, PRJNA337420 |

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
