## [Editor Report · eLife assessment]

This **valuable** study seeks to disentangle the different selective forces shaping the evolutionary dynamics of transposable elements (TEs) in the wild grass Brachypodium distachyon. Using haplotype-length metrics, and genetic and environmental differentiation tests, the authors present **convincing** evidence that positive selection on TE polymorphisms is rare and that the distribution of TE ages points to purifying selection being the main force acting on TE evolution in this species. This study will be relevant for anyone interested in the role of TEs in evolution and adaptation.

---

## [Referee Report · Reviewer #1 (Public Review)]

The study presented in this manuscript presents very convincing evidence that purifying selection is the main force shaping the landscape of TE polymorphisms in B. distachyon, with only a few putatively adaptive variants detected, even though most conclusions are based on the 10% of polymorphisms contributed by retrotransposons. That first conclusion is not novel, however, as it had already been clearly established in natural *A. thaliana* strains (Baduel et al. Genome Biol 2021) and in experimental D. simulans lines (Langmüller et al. NAR 2023). In contrast to the conclusions reached in A. thaliana, however, Horvath et al. report here a seemingly deleterious effect of TE insertions even very far away from genes (>5kb), a striking observation for a genome of relatively similar size. However, SNPs within these regions show similar allele frequency deviations, suggesting this effect may be due to mapping quality artefacts in gene poor regions of the genome. An additional caveat of this study is the lack of orthogonal benchmarking of the TE polymorphisms calls by a pipeline known for a high rate of false positives (see detailed Private Recommendations #1). The authors note that their conclusions are still valid using only the highest covered samples (>20x), yet this coverage threshold is relatively low and higher coverage would mostly reduce the rate of false negatives.

Nonetheless, this set of observations makes an important addition to the knowledge of TE dynamics in the wild and questions our understanding of the main molecular mechanisms through which TEs can impact fitness.

---

## [Referee Report · Reviewer #2 (Public Review)]

Transposable elements are known to have a strong potential to generate diversity and impact gene regulation, and they are thought to play an important role in plant adaptation to changing environments. Nevertheless, very few studies have performed genome-wide analyses to understand the global effect of selection on TEs in natural populations. Horvath et al., used available whole-genome re-sequencing data from a representative panel of B. distachyon accessions to detect TE insertion polymorphisms (TIPs) and estimate their time of origin. Using a thorough combination of population genomics approaches, the authors demonstrate that only a small amount of the TE polymorphisms are targeted by positive selection or potentially involved in adaptation. By comparing the age-adjusted population frequencies of TE polymorphisms and neutral SNPs, the authors found that retrotransposons are affected by purifying selection independently of their distance to genes. Finally, using forward simulations they were able to quantify the strength of selection acting on TE polymorphisms, finding that retrotransposons are mainly under moderate purifying selection, with only a minority of the insertions evolving neutrally.

Horvath et al., use a convincing set of strategies and their conclusions are well supported by the data. I think that incorporating polymorphism's age to the analysis of purifying selection is an interesting way to reduce the possible bias introduced by the fact that SNPs and TEs polymorphisms do not occur at the same pace. The fact that TE polymorphisms far from genes are also under purifying selection is an interesting result that reinforces the idea that trans-regulatory effect of TE insertions might not be a rare phenomenon, a matter that may be demonstrated in future studies.

---

## [Author Response]

The following is the authors’ response to the original reviews.

**eLife assessment**
This valuable study seeks to disentangle the different selective forces shaping the evolutionary dynamics of transposable elements (TEs) in the wild grass Brachypodium distachyon. Using haplotype-length metrics, and genetic and environmental differentiation tests, the authors present in large parts convincing evidence that positive selection on TE polymorphisms is rare, and that the distribution of TE ages points to purifying selection being the main force acting on TE evolution in this species. A caveat of this study, as of other studies that seek to assess TE insertion polymorphisms with short reads, is that the rates of false negatives and false positives are difficult to estimate, which may have major effects on the interpretation. This study will be relevant for anyone interested in the role of TEs in evolution and adaptation.

Thank you for considering our manuscript for publication in eLife. We appreciate the constructive comments and suggestions of the reviewers. We have addressed the raised issues by the reviewers. Below, we provide a more detailed response to each of the reviewer comments.

**Public Reviews:**

**Reviewer #1:**
The study presented in this manuscript presents very convincing evidence that purifying selection is the main force shaping the landscape of TE polymorphisms in B. distachyon, with only a few putatively adaptive variants detected, even though most conclusions are based on the 10% of polymorphisms contributed by retrotransposons. That first conclusion is not novel, however, as it had already been clearly established in natural *A. thaliana* strains (Baduel et al. Genome Biol 2021) and in experimental D. simulans lines (Langmüller et al. NAR 2023), two studies that the authors do not mention, or improperly mention. In contrast to the conclusions reached in A. thaliana, however, Horvath et al. report here a seemingly deleterious effect of TE insertions even very far away from genes (>5kb), a striking observation for a genome of relatively similar size. If confirmed, as a caveat of this study is the lack of benchmarking of the TE polymorphisms calls by a pipeline known for a high rate of false positives (see detailed Private Recommendations #1), this set of observations would make an important addition to the knowledge of TE dynamics in the wild and questioning our understanding of the main molecular mechanisms through which TEs can impact fitness.

Thank you for your positive evaluation of our paper. We have now adjusted the manuscript to include the mentioned studies (Line 330-333) and to address the issue of false positive and false negative calls. The detailed responses to all the raised points are below.

**Reviewer #2:**
Summary:Transposable elements are known to have a strong potential to generate diversity and impact gene regulation, and they are thought to play an important role in plant adaptation to changing environments. Nevertheless, very few studies have performed genome-wide analyses to understand the global effect of selection on TEs in natural populations. Horvath et al. used available whole-genome re-sequencing data from a representative panel of B. distachyon accessions to detect TE insertion polymorphisms (TIPs) and estimate their time of origin. Using a thorough combination of population genomics approaches, the authors demonstrate that only a small amount of the TE polymorphisms are targeted by positive selection or potentially involved in adaptation. By comparing the age-adjusted population frequencies of TE polymorphisms and neutral SNPs, the authors found that retrotransposons are affected by purifying selection independently of their distance to genes. Finally, using forward simulations they were able to quantify the strength of selection acting on TE polymorphisms, finding that retrotransposons are mainly under moderate purifying selection, with only a minority of the insertions evolving neutrally.Strengths:Horvath et al., use a convincing set of strategies, and their conclusions are well supported by the data. I think that incorporating polymorphism's age into the analysis of purifying selection is an interesting way to reduce the possible bias introduced by the fact that SNPs and TEs polymorphisms do not occur at the same pace. The fact that TE polymorphisms far from genes are also under purifying selection is an interesting result that reinforces the idea that the trans-regulatory effect of TE insertions might not be a rare phenomenon, a matter that may be demonstrated in future studies.Weaknesses:TEs from different classes and orders strongly differ in multiple features such as size, the potential impact of close genes upon insertion, insertion/elimination ratio (ie, MITE/TIR excision, solo-LTR formation), or insertion preference. Given such diversity, it is expected that their survival rates on the genome and the strength of selection acting on them could be different. The authors differentiate DNA transposons and retrotransposons in some of the analyses, the specificities of the most abundant plant TE types (ie, LTR/Gypsy, LTR/Copia, MITE DNA transposons) are not considered.The authors used a short-read-based approach to detect TIPs and TAPs. It is known that detecting TE polymorphisms is challenging and can lead to false negatives, depending on the method used and the sequencing coverage. The methodology used here (TEPID) has been previously applied to other species, but it is unclear if the sensitivity of the TIP/TAP caller is equivalent to that of the SNP caller and how these potential differences may affect the results.

Thank you for your positive evaluation of our paper. We have now adjusted the manuscript and the discussion to include the mentioned points on the different TE superfamilies and the reliability of the TE calls. The detailed responses to all the raised points are below.

**Private Recommendations:**

**Reviewer #1:**
(1) TE polymorphisms (presence and absence variants) were called from short-read sequencing data using a pipeline (TEPID, Stuart et al. eLife 2016) that is known to have a low specificity as well as a low sensitivity in its detection of presence variants (Baduel et al. MIMB 2021). An assessment of the rate of false positives and false negatives in the data presented in this study and how it varies across TE superfamilies is therefore of crucial importance as it may bias all downstream analyses, especially if it impacts the identification of polymorphisms contributed by retrotransposons, as these are the basis of most conclusions of the manuscript. Nonetheless, the fact that the PCA of the polymorphisms contributed by DNA transposons is less able to distinguish genetic clades than with those contributed by retrotransposons, suggests the issue of false positives is most preeminent for DNA transposons. However, high rates of false positives may explain why no significant increase in TE frequency is detected within selective sweep regions, a result that runs against the expectation of hitch-hiking of neutral or weakly deleterious polymorphisms which the authors claim is the category of many TE polymorphisms. Furthermore, given that the reference genome belongs to the B_east clade, and the TEPID is better at calling absence than presence it may bias analyses in this clade (where clade-specific insertions will take the form of absence in other clades which are well detected) compared to other clades (where clade-specific insertions will be presence polymorphisms and may be missed). A benchmark of TE polymorphism calls could be done by de novo assembling one genome from each clade or by cross-checking at least the presence variant calls from TEPID with those made with another of the many TE calling pipelines available.

We agree with this issue raised by both reviewers regarding the effects of false negative and false positive TE calls. We also think that some reasonable follow-ups should be done to check the potential impact of the false negative and false positive TE calls on the presented results, without turning the manuscript in a method comparison paper as this is not the main goal of this study. Therefore, we generated a subsample of our dataset that included only accession with an average genome wide mapping coverages of at least 20x, as the false negative TE call rate is correlated with the mapping coverage and a high mapping coverage is expected to lead to a reduction in the false negative TE call rates. We then used this subsample to check if our results would change if our dataset had a lower false negative TE call rate. However, reducing the rate of false negative calls through the use of only higher coverage samples did not change our results and interpretations.

Re-running the ANCOVA analyses revealed similar results regarding the accumulation of TEs in selective sweep regions. This was added to the main text Line 143-148: “Similar results were obtained when investigating the number of fixed TE polymorphisms (Additional file 2: Table S1) and the allele frequency of TE polymorphisms (Additional file 2: Table S2) in high iHS regions using a subset of our dataset with an expected lower false negative TE call rate, that only included samples with a genome-wide mapping coverage of at least 20x (see Discussion and Materials and Methods for more details).” and in Additional file 2: Table S1 and S2.

Further, we re-ran the age-adjusted SFS based on this subset of our dataset and found that the results and conclusions from the age-adjusted SFS were not only driven by false negative TE calls. This was also included in the text Line 338-349: “One caveat of the approach used in this study is that TE calling pipelines based on short-reads tend to have higher false positive and false negative call rates than SNP calling pipelines, which is also the case for the TEPID TE calling pipeline used here [57, 59]. A high false negative TE calling rate however might bias our TE frequency estimates toward lower frequencies, which could drive the observed patterns in the age-adjusted SFS. To assess if the false negative TE calling rate in our study substantially affected our results, we re-run the age-adjusted SFS on a subset of our dataset only including samples with a genome-wide mapping coverage of at least 20x, as higher mapping coverages are expected to reduce the false negative call rate [27, 59]. Using the TE allele frequencies estimated based on this subset of our data to estimate Δ frequency revealed similar results of the age-adjusted SFS based on the whole dataset (Additional file 1: Fig. S9), indicating that our observation of retrotransposons evolving under purifying selection is not solely driven by a high false negative TE calling rate.” and in Additional file 1: Fig. S9.

The details of this analyses have been added to the materials and methods Line 493-498: “Mapping coverage is known to influence false discovery rate [27, 59]. To investigate the impact of false positive and false negative TE calls on our results, we down sampled the TE dataset to only include TEs that have been called in samples that had at least an average mapping coverage of 20x. The allele frequencies of TEs present in our high coverage dataset was recalculated only considering samples with at least an average mapping coverage of 20x. This second TE dataset was then used to check if using a dataset with a higher mapping coverage and presumably a lower false TE calling rate impacted our results.”

(2) If confirmed, the observation that retrotransposons located more than 5kb away from genes appear to be also affected by purifying selection (L209) is indeed surprising. The authors should add a comparison with SNPs at the same distance from genes to strengthen the claim and make sure it is not the result of mapping artifacts, such as alignment quality dropping far away from genes.

We added a comparison of the age-adjusted SFS of SNPs and retrotransposons more than 5 kb away from genes to evaluate if the observed shape of the age-adjusted SFS of retrotransposons more than 5 kb away from genes were due to artefacts. The results are included on line 383-389: “Finally, we tested whether TE polymorphisms located more than 5 kb away from genes are evolving under purifying selection could be due to mapping or other artefacts by comparing the shape of the age-adjusted SFS of retrotransposons and SNPs more than 5 kb away from genes. However, the age-adjusted SFS of SNPs 5 kb away from genes differs from the one of retrotransposons (Additional file 1: Fig. S10), indicating that the shape of the age-adjusted SFS of retrotransposons more than 5 kb away from genes is not likely to be the result of artefacts in regions of the genome far away from genes.” and Additional file 1: Fig. S10.

(3) The authors' claim that most TE polymorphisms are under weak to moderate purifying selection (L273) relies on the comparison of the age of polymorphisms in the oldest age bin with forward simulations. However, the conclusions from these comparisons cannot be extrapolated to the fitness effects of all TE polymorphisms as variants in the oldest age bin are de facto a biased sample of the variants of a category, a point the authors highlight.

We adjusted the mentioned paragraph to better highlight this point. Line 390-397: “To further ascertain the strength of purifying selection, we used forward simulation and showed that simulations assuming a moderately weak selection pressure (S = -5 or S = -8) against TE polymorphisms best fitted our observed data. In theory, no TE polymorphisms under strong purifying selection should be present in a natural population, as such mutations are expected to be quickly lost, especially in a predominantly selfing species where most loci are expected to be homozygous. Therefore, it is not surprising that TE polymorphisms which persist in B. distachyon are under weak to moderate selection, as also shown, for example, for the L1 retrotransposons in humans [27] or the BS retrotransposon family in *Drosophila melanogaster* [62].”

L220-228 for high-effect SNPs. Indeed, the most deleterious TE polymorphisms would be purged very quickly and never contribute to variants in the oldest age bin. Unless new arguments can be made to support this claim, this conclusion should be rephrased to claim instead that even the oldest TE polymorphisms are still mostly non-neutral and under weak to moderate purifying.

This has been adjusted. Line 231-232: “. Hence, even the oldest retrotransposon polymorphisms seem to be mostly non-neutral and are affected by purifying selection.”

L214: replace smaller with more negative for clarity.

Done.

L233: Given the discussion L220-228, the oldest age bin seems to be biased in its composition and thus not useful for comparisons. The sentence should therefore be rephrased to reflect that DNA transposon polymorphisms appear to be actually less deleterious than high-effect SNPs in S9A and B based on the penultimate age bin.

This has been fixed.

**Reviewer #2:**
I wonder if false negative detection could artificially increase the evidence for purifying selection by increasing the amount of low-frequency variants. This could be easily checked if long-read data or genome assembly is available for any of the samples in the collection, by comparing the TIP/TAP prediction with the actual sequence.

We agree with this point from the reviewers that false negative calls can lead to misinterpretations of the observed low-frequencies of the TEs. (But see response to the first comment of reviewer #1). Unfortunately, long-read data from the sample used here are not available to estimate false negative call rates. However, to check if the observed results are manly driven by high false negative rates, we re-run the age-adjusted SFS based on samples with at least 20x mapping coverage, which should result in the reduction the false negative TE calling rate. The results and conclusions from this second analyses were included in the text Line 338-349: “One caveat of the approach used in this study is that TE calling pipelines based on short-reads tend to have higher false positive and false negative call rates than SNP calling pipelines, which is also the case for the TEPID TE calling pipeline used here [57, 59]. A high false negative TE calling rate however might bias our TE frequency estimates toward lower frequencies, which could drive the observed patterns in the age-adjusted SFS. To assess if the false negative TE calling rate in our study substantially affected our results, we re-run the age-adjusted SFS on a subset of our dataset only including samples with a genome-wide mapping coverage of at least 20x, as higher mapping coverages are expected to reduce the false negative call rate [27, 59]. Using the TE allele frequencies estimated based on this subset of our data to estimate Δ frequency revealed similar results of the age-adjusted SFS based on the whole dataset (Additional file 1: Fig. S9), indicating that our observation of retrotransposons evolving under purifying selection is not solely driven by a high false negative TE calling rate.” and in Additional file 1: Fig. S9.

Supplementary Figure S1. DNA transposons are much worse at separating the samples in comparison to LTR-retrotransposons. Doesn´t this suggest that these two classes have very different dynamics in the population and maybe different intensities of the selection forces acting on them? Could this profile be explained as DNA transposons being older and likely more fixed in all the clades, whereas retrotransposons are more recent and more specific to some populations? Another possibility might be that some B. distachyon DNA transposons had an unusually high excision rate. In any case, in my opinion, this reinforces the need to study the different TE orders in more detail.

Indeed, different TE orders and superfamilies can have different excision rates, age distributions and be under different selective regimes. To investigate the possibility that different TE orders are affected by very different selective regimes, we split our TE dataset into the four different TE types: Copia, Ty3, Helitron and MITE. We than re-run the age-adjusted SFS analyses and added our results to the text Line 422-430: “To further examine our conclusion on purifying selection, we investigated the selective regime affecting different retrotransposons and DNA-transposons superfamilies. Thereby, we generated age-adjusted SFS for the four most common TE superfamilies Copia, Ty3 (also known under the name Gypsy, but we will avoid using this name because of its problematic nature see [71]), Helitron and MITE and found similar deviations of the Δ frequency from 0 in the four investigated TE superfamilies (Additional file 1: Fig. S12–S15). These results indicate that our conclusion on the broad effect of purifying selection is not driven by a single TE superfamily but is at least common among the four most numerous TE superfamilies.” and in Additional file 1: Fig. S12- S15.

Line 112: "most TE polymorphisms in our dataset were young and only a few were very old". Does this change substantially among TE orders/superfamilies?

Indeed, there are some differences in the age distribution of the TEs depending on the superfamilies, However, the differences are no substantial as the age bins in the age-adjusted SFS of the different TE superfamilies are fairly similar. See Additional file 1: Fig. S12-S15.

Figure 2. Is difficult to read, especially lower panels. I think the grey border of the boxplots makes visualization difficult.

The gray borders have been removed.